# A molecular mechanism for LINC complex branching by structurally diverse SUN-KASH 6:6 assemblies

Manickam Gurusaran[1,2], Owen Richard Davies[1,2]*

[1]Institute of Cell Biology, University of Edinburgh, Edinburgh, United Kingdom; [2]Biosciences Institute, Faculty of Medical Sciences, Newcastle University, Newcastle, United Kingdom

**Abstract** The Linker of Nucleoskeleton and Cytoskeleton (LINC) complex mechanically couples cytoskeletal and nuclear components across the nuclear envelope to fulfil a myriad of cellular functions, including nuclear shape and positioning, hearing, and meiotic chromosome movements. The canonical model is that 3:3 interactions between SUN and KASH proteins underlie the nucleocytoskeletal linkages provided by the LINC complex. Here, we provide crystallographic and biophysical evidence that SUN-KASH is a constitutive 6:6 complex in which two constituent 3:3 complexes interact head-to-head. A common SUN-KASH topology is achieved through structurally diverse 6:6 interaction mechanisms by distinct KASH proteins, including zinc-coordination by Nesprin-4. The SUN-KASH 6:6 interface provides a molecular mechanism for the establishment of integrative and distributive connections between 3:3 structures within a branched LINC complex network. In this model, SUN-KASH 6:6 complexes act as nodes for force distribution and integration between adjacent SUN and KASH molecules, enabling the coordinated transduction of large forces across the nuclear envelope.

*For correspondence:
owen.davies@ed.ac.uk

Competing interests: The authors declare that no competing interests exist.

## Introduction

The nuclear envelope partitions nuclear components from the cytoskeleton, thereby necessitating their mechanical coupling across the nuclear envelope to enable cytoskeletal function in the structure and positioning of nuclear contents. This is achieved by the Linker of Nucleoskeleton and Cytoskeleton (LINC) complex, which traverses the nuclear envelope and binds to cytoskeletal and nuclear structures to mediate force transduction between these partitioned components (*Haque et al., 2006*; *Crisp et al., 2006*; *Lee and Burke, 2018*; *Meinke and Schirmer, 2015*; *Figure 1a*). In this capacity, the LINC complex is essential for cellular life, performing critical functions in nuclear structure, shape, and positioning (*Alam et al., 2015*; *Luxton et al., 2010*; *Crisp et al., 2006*), in addition to tissue-specific functions including sound perception in the inner ear and chromosome movements during meiosis (*Horn et al., 2013a*; *Roux et al., 2009*; *Horn et al., 2013b*; *Lee et al., 2015*). Further, mutations of the LINC complex and its interacting partners are associated with human laminopathies, including Hutchison-Gilford progeria syndrome and Emery-Dreifuss muscular dystrophy (*Meinke et al., 2011*; *Chen et al., 2012*; *Taranum et al., 2012*; *Zhou et al., 2018b*; *Chang et al., 2019*).

The LINC complex is formed of SUN (Sad1 and UNC84 homology) domain and KASH (Klarsicht, ANC-1, and Syne homology) domain proteins (*Padmakumar et al., 2005*; *Haque et al., 2006*; *Crisp et al., 2006*), which interact immediately below the outer nuclear membrane, through complex formation between their C-terminal eponymous SUN and KASH domains (*Sosa et al., 2012*; *Wang et al., 2012*; *Zhou et al., 2012*). SUN proteins then traverse the approximately 50 nm perinuclear space and cross the inner nuclear membrane, enabling their N-termini to bind to nuclear

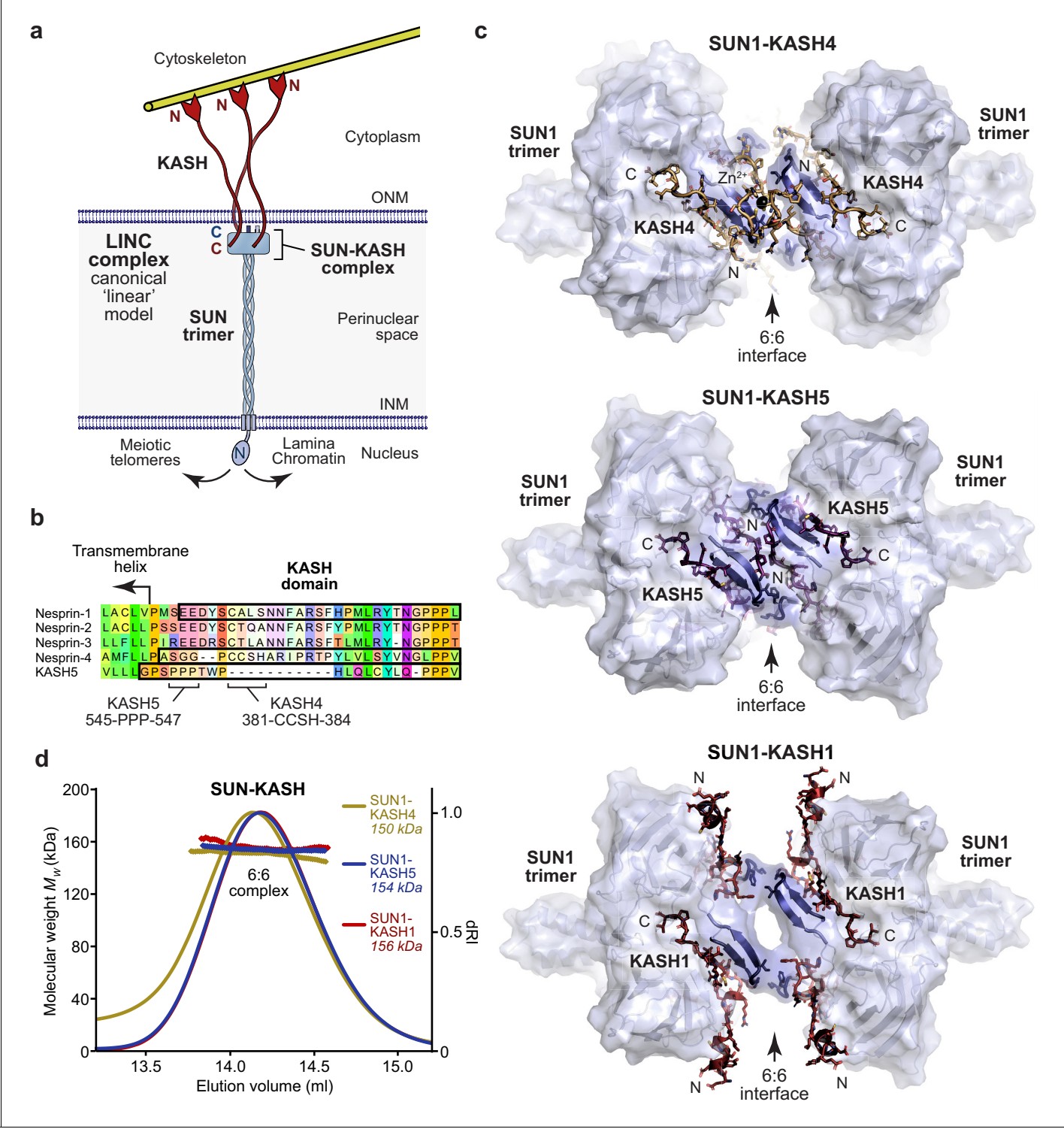

**Figure 1.** SUN-KASH complexes are 6:6 head-to-head assemblies. (**a**) The Linker of Nucleoskeleton and Cytoskeleton (LINC) complex traverses the nuclear envelope to transmit forces between the cytoskeleton and nuclear components. The canonical model of the LINC complex is a linear structure formed of SUN and Nesprin proteins, which interact via a 3:3 complex between their SUN and KASH domains within the peri-nuclear space, and cross the inner and outer nuclear membranes (INM and ONM), respectively. (**b**) Sequence alignment of the KASH domains of human Nesprins 1–4 and KASH5. In this study, KASH1, KASH4, and KASH5 refer to the C-terminal KASH domains of Nesprin-1, Nesprin-4, and KASH5, respectively, which are highlighted (black outline), and key amino acids within KASH4 and KASH5 are indicated. (**c**) Crystal structures of human SUN1-KASH4 (top), SUN1-KASH5 (middle), and SUN1-KASH1 (bottom). The SUN1 molecular surface is displayed with SUN1 KASH-lids highlighted in blue as cartoons, and KASH

*Figure 1 continued on next page*

*Figure 1 continued*

sequences are represented as sticks (yellow, purple, and red, respectively). All structures are 6:6 complexes in which KASH proteins lie at the midline head-to-head interface between SUN1 trimers. (**d**) SEC-MALS analysis showing differential refractive index (dRI) profiles with fitted molecular weights (*Mw*) plotted as diamonds across elution peaks. SUN1-KASH4, SUN1-KASH5, and SUN1-KASH1 form 6:6 complexes in solution, with experimental molecular weights of 150, 154, and 156 kDa, respectively (theoretical 6:6 – 155, 155, and 157 kDa). Representative of more than three replicates using different protein preparations. Full elution profiles are shown in *Figure 1—figure supplement 2*.

The online version of this article includes the following figure supplement(s) for figure 1:

**Figure supplement 1.** Crystal structures of SUN-KASH complexes.
**Figure supplement 2.** SUN-KASH complexes are 6:6 assemblies.

contents, including reported interactions with the nuclear lamina (*Crisp et al., 2006*; *Haque et al., 2006*; *Haque et al., 2010*), chromatin (*Chi et al., 2007*), and the telomeric ends of meiotic chromosomes (*Shibuya et al., 2014*). Similarly, KASH domain proteins cross the outer nuclear membrane and have large cytoplasmic extensions to enable their N-termini to bind to the cytoskeleton (*Spindler et al., 2019*; *Starr and Fridolfsson, 2010*). Thus, the LINC complex axis is established by a peri-nuclear SUN-KASH core interaction and mechanically couples the cytoskeleton and nuclear contents (*Figure 1a*).

In mammals, there are five SUN proteins, of which SUN1 and SUN2 are widely expressed and perform partially redundant functions (*Lei et al., 2009*; *Zhang et al., 2009*). There are similarly multiple KASH proteins, four of which are Nesprins (Nuclear Envelope Spectrin Repeat proteins). Nesprin-1 and Nesprin-2 are widely expressed, perform overlapping functions and contain large cytoplasmic spectrin-repeat domains with N-termini that bind to actin (*Banerjee et al., 2014*; *Sakamoto et al., 2017*; *Zhou et al., 2018a*). Nesprin-3 shares a similar KASH domain but its cytoplasmic region binds to plectin, mediating interactions with intermediate filaments (*Wilhelmsen et al., 2005*). The two most divergent KASH proteins, Nesprin-4 and KASH5, exhibit substantial sequence diversity within their KASH domains (*Figure 1b*). Nesprin-4 functions in the outer hair cells of the inner ear and is essential for hearing (*Horn et al., 2013a*). Its N-terminus interacts with kinesin, which mediates microtubule binding and plus-end directed movements that achieve the basal positioning of nuclei (*Horn et al., 2013a*; *Roux et al., 2009*). KASH5 functions in meiosis and is essential for fertility (*Horn et al., 2013b*; *Morimoto et al., 2012*). Its N-terminus interacts with dynein-dynactin (*Morimoto et al., 2012*; *Horn et al., 2013b*), which mediates microtubule binding and minus-end directed motility that drives rapid chromosomal movements to facilitate homologous chromosome pairing (*Lee et al., 2015*; *Zetka et al., 2020*). Thus, KASH proteins execute a range of LINC complex functions in transmitting actin forces, plus-/minus-end directed microtubule movements and the tensile strength of intermediate filaments into the nucleus.

The canonical model of the LINC complex is based on crystal structures of the SUN-KASH domain complexes formed between SUN2 and Nesprin-1/2 (*Sosa et al., 2012*; *Wang et al., 2012*). The SUN domain adopts a 'three-leaf clover'-like structure, in which a globular trimer extends from a short N-terminal trimeric coiled-coil (*Sosa et al., 2013*). KASH domains are intertwined between SUN protomers and their path is defined by three distinct regions. The KASH C-terminus contains a triple proline motif that packs between the globular cores of SUN protomers. The KASH mid-region winds around the trimeric arc and is wedged between the globular core of one SUN protomer and a β-turn-β loop, known as the KASH-lid, of the adjacent protomer. The KASH N-terminus then turns by >90° to radiate out from the trimer axis and forms a disulphide bond with a SUN protomer (between SUN2 and KASH1 amino acids C563 and C8774, respectively), which is predicted to enhance the tensile strength of SUN-KASH (*Jahed et al., 2015*; *Sosa et al., 2012*). The extensive 3:3 complex of three KASH domains bound to a single SUN trimer was interpreted as the biological unit of the crystal lattice (*Sosa et al., 2012*; *Wang et al., 2012*). On this basis, it was proposed that the LINC complex consists of a SUN-KASH 3:3 complex that is orientated vertically to allow KASH proteins to cross the outer nuclear membrane and SUN to form an extended trimeric coiled-coil that spans the peri-nuclear space (*Sosa et al., 2012*; *Sosa et al., 2013*; *Figure 1a*). In support of this model, the luminal region of SUN2 was shown to be trimeric in vitro by analytical ultracentrifugation, SEC-MALS, and gel filtration (*Sosa et al., 2012*; *Nie et al., 2016*; *Zhou et al., 2012*; *Jahed et al., 2018b*) and upon targeting to the nuclear envelope in vivo, the luminal regions of SUN2 and SUN1 were shown to form trimers and larger structures by fluorescence fluctuation spectroscopy

(*Hennen et al., 2017*; *Hennen et al., 2018*). However, the stoichiometry of SUN-KASH complexes has not yet been tested in solution. Further, whilst it has been widely recognised that branching or higher order assembly of LINC complexes may be advantageous in distributing large forces and achieving coordinated motions (*Zhou et al., 2012*; *Lu et al., 2008*; *Jahed et al., 2018a*; *Wang et al., 2012*; *Sosa et al., 2013*; *Lu et al., 2012*), we have hitherto lacked structural evidence and a molecular basis for higher order assembly of the LINC complex.

Here, we provide crystallographic and biophysical evidence in support of the LINC complex forming a branched network. We find that SUN-KASH complexes between SUN proteins and Nesprin-4, KASH5 and Nesprin-1 are 6:6 structures formed of constitutive interactions between two 3:3 complexes. The three distinct KASH domains provide structurally diverse but related 6:6 interfaces that achieve the same topology with potential hinge-like motion between SUN trimers. The SUN-KASH 6:6 interface consists of a 'head-to-head' interaction between SUN's trimeric C-terminal 'heads', thereby providing a mechanistic basis for formation of a branched LINC complex network. Thus, we propose that SUN-KASH domain complexes act as nodes for branching and integration between LINC complexes to achieve the coordinated transduction of large forces across the nuclear envelope.

## Results

### SUN1-KASH complexes are 6:6 hetero-oligomers

The previously reported crystal structures of SUN-KASH complexes between SUN2 and Nesprins 1–2 revealed almost identical structures that were interpreted as 3:3 hetero-oligomers (*Sosa et al., 2012*; *Wang et al., 2012*). The KASH domains of Nesprin-4 and KASH5 exhibit sequence divergence from Nesprins 1–3, including the presence of N-terminal motifs of 381-CCSH-384 and 545-PPP-547, which are conserved within Nesprin-4 and KASH5 sequences, respectively (*Figure 1b*). On this basis, we reasoned that Nesprin-4 and KASH5 may impose unique SUN-KASH structures that differ from the classical architecture of Nesprin 1–3 complexes, which may underlie their specialised functional roles. We thus solved the X-ray crystal structures of SUN-KASH complexes formed between the SUN domain of SUN1 and KASH domains of Nesprin-4 and KASH5 (herein referred to as SUN1-KASH4 and SUN1-KASH5). The SUN1-KASH4 structure was solved at a resolution of 2.75 Å and revealed a 6:6 assembly in which two globular 3:3 complexes are held in a head-to-head configuration through zinc-coordination by opposing KASH4 molecules across the 6:6 interface (*Figure 1c*, *Table 1* and *Figure 1—figure supplement 1a,b*). The SUN1-KASH5 crystal structure was solved at 1.54 Å resolution and revealed a similar 6:6 assembly in which opposing 3:3 complexes are held together by extensive interactions between opposing KASH5 molecules and KASH-lids (*Figure 1c*, *Table 1* and *Figure 1—figure supplement 1a,c*). Thus, both Nesprin-4 and KASH5 form SUN-KASH 6:6 hetero-oligomers in which similar topologies of head-to-head 3:3 complexes are achieved through structurally diverse 6:6 interfaces.

Is the 6:6 assembly unique to SUN-KASH complexes formed by Nesprin-4 and KASH5? We next solved the crystal structure of the SUN-KASH complex between SUN1 and Nesprin-1 (herein referred to as SUN1-KASH1). The SUN1-KASH1 structure was solved at 1.82 Å resolution and demonstrated a similar 6:6 head-to-head assembly, albeit with less extensive interface-spanning interactions provided solely by opposing KASH-lids (*Figure 1c*, *Table 1* and *Figure 1—figure supplement 1a,d*). The electron density indicated the presence of a molecule bound close to the 6:6 interface, which we interpreted as a disordered HEPES molecule from the crystallisation condition. This likely provided structural rigidity that underlies the high resolution of the dataset, but was not essential for the structure as we solved numerous other datasets at lower resolution in which an identical 6:6 interface was present in absence of a bound molecule (data not shown). Importantly, the SUN1-KASH1 structure closely matches the previous SUN2-KASH1/2 structures, in which similar 6:6 interfaces were present in the crystal lattice but were thought to be crystal contacts (*Figure 1—figure supplement 1e*; *Sosa et al., 2012*; *Wang et al., 2012*). It was thus critical to determine whether SUN1-KASH1 is a 6:6 complex in solution. We utilised size-exclusion chromatography multi-angle light scattering (SEC-MALS) as the gold standard for determining molecular species. SEC-MALS revealed that all SUN1-KASH complexes exist solely as 6:6 hetero-oligomers (*Figures 1d* and *2a*, and *Figure 1—figure supplement 2*). Moreover, their 6:6 complexes remained intact at the lowest detectable

**Table 1.** Data collection, phasing, and refinement statistics.

| | Sun1-kash4 | Sun1-kash5 | Sun1-kash1 |
|---|---|---|---|
| PDB accession | 6R16 | 6R2I | 6R15 |
| Data collection | | | |
| Space group | $P2_12_12_1$ | $P6_322$ | $P6_322$ |
| Cell dimensions | | | |
| $a, b, c$ (Å) | 104.37, 117.21, 138.42 | 80.16, 80.16, 177.62 | 80.45, 80.45, 182.55 |
| $\alpha, \beta, \gamma$ (°) | 90.00, 90.00, 90.00 | 90.00, 90.00, 120.00 | 90.00, 90.00, 120.00 |
| Wavelength (Å) | 0.9795 | 0.9282 | 0.9282 |
| Resolution (Å) | 48.83–2.75 (2.85–2.75)* | 88.81–1.54 (1.57–1.54)* | 65.09–1.82 (1.87–1.82)* |
| $R_{meas}$ | 0.111 (1.355) | 0.070 (1.551) | 0.085 (2.192) |
| $R_{pim}$ | 0.056 (0.741) | 0.015 (0.329) | 0.019 (0.465) |
| Completeness (%) | 99.7 (97.5) | 97.5 (100.0) | 100.0 (100.0) |
| $I/\sigma(I)$ | 15.4 (1.4) | 23.5 (2.2) | 21.5 (1.7) |
| $CC_{1/2}$ | 0.999 (0.488) | 1.000 (0.801) | 1.000 (0.776) |
| Multiplicity | 7.1 (5.6) | 21.3 (22.1) | 20.6 (21.9) |
| | | | |
| Refinement | | | |
| Resolution (Å) | 47.67–2.75 | 23.99–1.54 | 65.09–1.82 |
| No. reflections | 44658 | 49372 | 32230 |
| $R_{work}$ / $R_{free}$ | 0.2190/0.2549 | 0.1495/0.1683 | 0.1587/0.1817 |
| Cruickshank DPI (Å) | 0.25 | 0.06 | 0.06 |
| No. atoms | 10562 | 2127 | 2107 |
| Protein | 10451 | 1817 | 1845 |
| Ligand/ion | 21 | 1 | 26 |
| Water | 90 | 309 | 236 |
| $B$ factors | 80.64 | 36.45 | 48.87 |
| Protein | 80.87 | 35.09 | 47.73 |
| Ligand/ion | 68.03 | 18.87 | 119.12 |
| Water | 56.37 | 44.50 | 50.06 |
| R.m.s. deviations | | | |
| Bond lengths (Å) | 0.002 | 0.011 | 0.013 |
| Bond angles (°) | 0.444 | 1.076 | 0.995 |

\* Values in parentheses are for highest-resolution shell.

concentrations (*Figure 2b–d*) and we failed to detect 3:3 complexes in any biochemical conditions tested. Thus, we conclude that the SUN-KASH complexes formed by SUN1 are constitutive 6:6 hetero-oligomers in which two 3:3 structures are locked in head-to-head interactions. Hence, their 6:6 interfaces could mediate the physical coupling of adjacent LINC complexes within the peri-nuclear space.

## Structural diversity within the SUN1-KASH 6:6 interface

Our SUN1-KASH crystal structures reveal the formation of similar 6:6 architectures through diverse head-to-head interfaces. Whilst the C-termini of all three KASH domains adopt the same structure, their N-termini differ substantially (*Figure 3a,b*). KASH1 undergoes a turn of >90° to radiate from the trimer axis, similar to the previously reported SUN2-KASH1/2 structures (*Figure 1—figure supplement 1e*), whereas KASH4 and KASH5 follow the arc of the SUN1 trimer, enabling them to contribute directly to the 6:6 interface (*Figure 3a,b*).

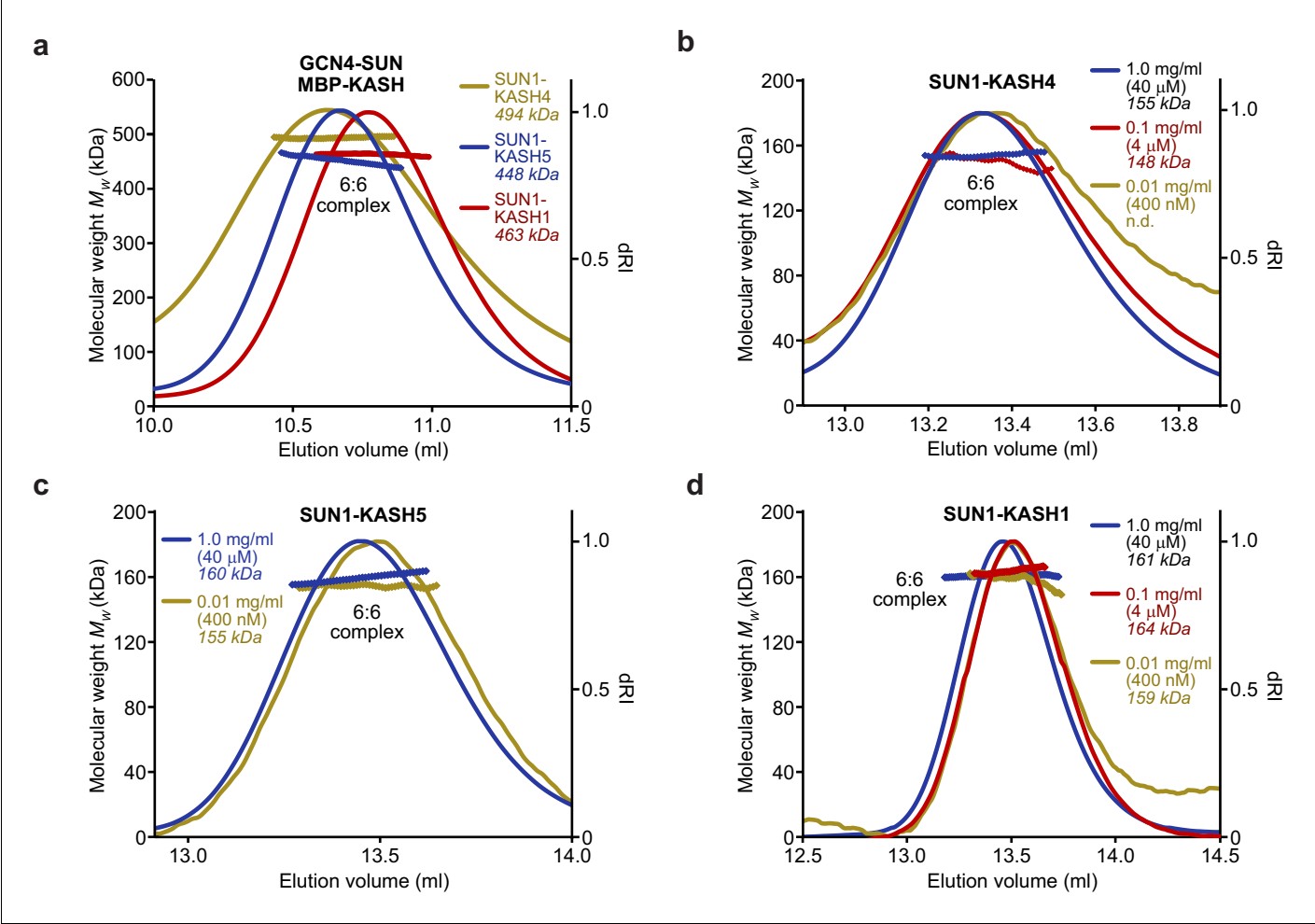

**Figure 2.** SUN-KASH 6:6 complexes are stable in solution. (a–d) SEC-MALS analysis performed in 20 mM Tris pH 8.0, 150 mM KCl, 2 mM DTT. (a) GCN4-SUN1 and MBP-KASH form 6:6 complexes of 494 kDa (KASH4, yellow), 448 kDa (KASH5, blue), and 463 kDa (KASH1, red) (theoretical 6:6 – 464, 464, and 466 kDa). (b–d) Dilution series of SUN-KASH complexes analysed at 1.0 mg/ml (blue), 0.1 mg/ml (red), and 0.01 mg/ml (yellow) for (b) SUN1-KASH4 (theoretical 6:6 – 155 kDa), (c) SUN1-KASH5 (theoretical 6:6 – 155 kDa), and (d) SUN1-KASH1 (theoretical 6:6 – 157 kDa).

SUN1-KASH4 adopts an unusual conformation in which the 6:6 complex is held together by three interface spanning zinc-sites, each coordinated by opposing KASH4 molecules (*Figure 3c* and *Figure 3—figure supplement 1a*). The presence of metal ions in the crystal structure was confirmed by corresponding peaks in anomalous difference electron density maps (*Figure 3d*), and their identity as zinc ions that were co-purified from bacterial expression was confirmed by the spectrophotometric determination of three zinc ions per 6:6 complex in solution that were lost upon pre-incubation with EDTA (*Figure 3e*). The zinc-sites are coordinated by asymmetric ligands from 381-CCSH-384 motifs of opposing KASH4 molecules, comprising C381 and C382 from one molecule, and C382 and H384 from the other (*Figure 3c*), and mutation of both cysteine residues to serine was sufficient to preclude zinc-binding (*Figure 3e*). The three zinc-sites form a tripod of interactions that provide the sole interface-spanning contacts between opposing 3:3 complexes (*Figure 3b*).

SUN1-KASH5 demonstrates the most extensive 6:6 interface in which KASH5 molecules and SUN1 KASH-lids from opposing 3:3 complexes wind around each other in a right-handed screw to create a complete circumferential interface enclosing a hollow core, similar to a β-barrel fold (*Figure 3f* and *Figure 3—figure supplement 1b*). KASH5 follows an almost linear path, packed between a SUN1 globular core and KASH-lids of opposing SUN1 protomers, with N-terminal 545-PPP-547 motifs of opposing molecules interacting across the interface. KASH5 and KASH4 follow similar paths, with KASH5 PPP-motif interactions and KASH4 zinc-sites located at the same positions

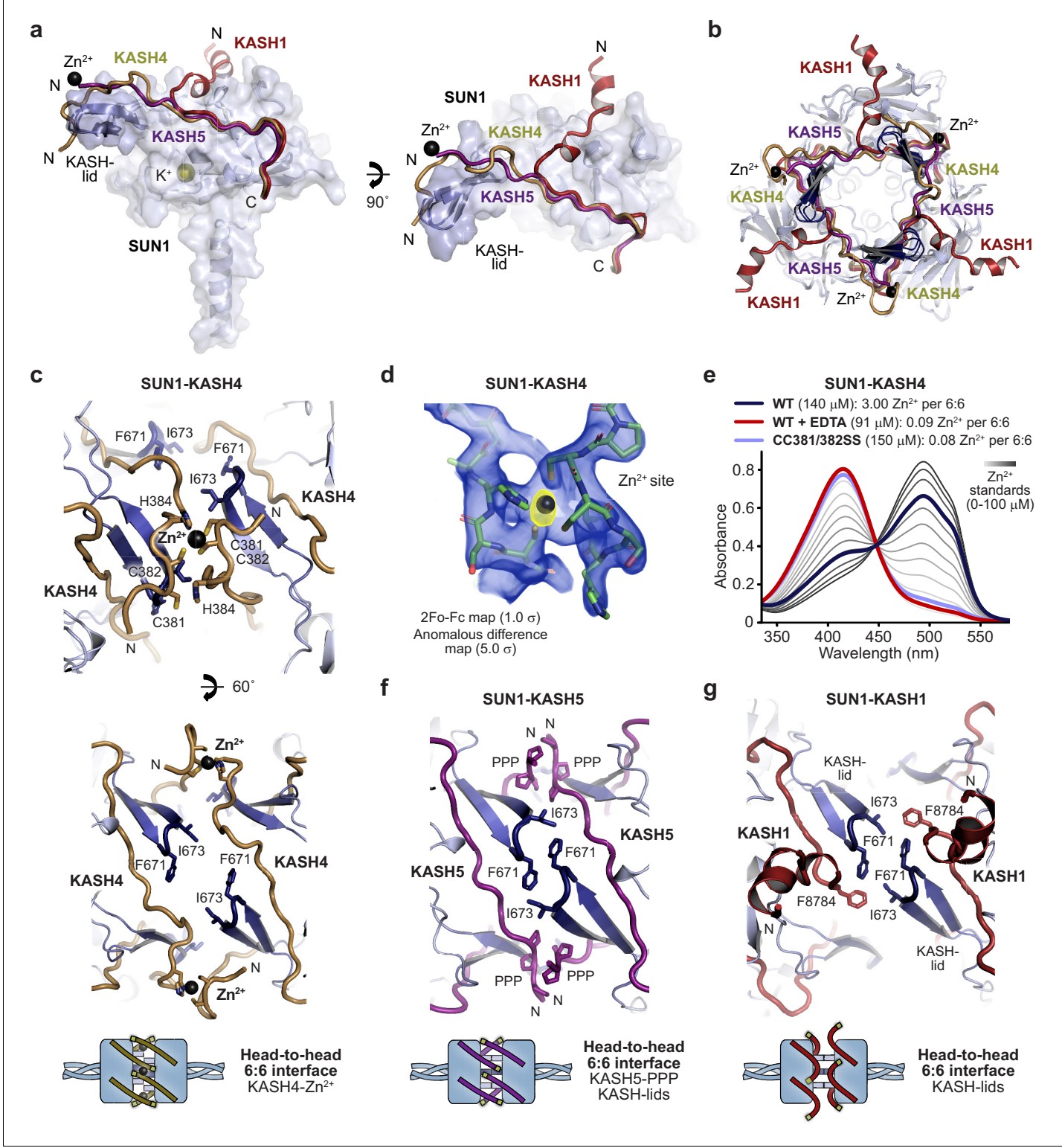

**Figure 3.** Specialised KASH sequences provide distinct SUN-KASH 6:6 assembly mechanisms. (**a**) SUN-KASH 1:1 protomers from SUN1-KASH4, SUN1-KASH5, and SUN1-KASH1 crystal structures, superposed, and displayed as the SUN1 molecular surface with KASH-lids highlighted in blue as cartoons, and KASH sequences represented as cartoons (yellow, purple, and red, respectively). (**b**) Cross-section through the head-to-head interface of superposed SUN1-KASH4, SUN1-KASH5, and SUN1-KASH1 6:6 assemblies such that their constituent 3:3 complexes are visible. (**c**) Structural details of the SUN1-KASH4 6:6 interface, showing a zinc-binding site in which opposing KASH4 chains provide asymmetric ligands C381 and C382, and C382 and H384 (top), and the lack of interface-spanning interactions between opposing SUN1 KASH-lids (bottom). (**d**) 2Fo-Fc (blue) and anomalous difference

*Figure 3 continued on next page*

*Figure 3 continued*

(yellow) electron density maps contoured at 1.0 σ and 5.0 σ, respectively, at a zinc-binding site of SUN1-KASH4. (**e**) Spectrophotometric determination of zinc content for SUN1-KASH4 wild-type (dark blue; 3.00 $Zn^{2+}$ per 6:6), wild-type with EDTA treatment prior to gel filtration (red; 0.09 $Zn^{2+}$ per 6:6), and CC381/382SS (light blue; 0.08 $Zn^{2+}$ per 6:6), using metallochromic indicator PAR, with zinc standards shown in a gradient from light to dark grey (0–100 μM). Representative of three replicates. (**f**) Structural details of the SUN1-KASH5 6:6 interface, demonstrating interface-spanning interactions between PPP-motifs (amino acids 545-PPP-547) of opposing KASH5 chains, and between amino acids F671 and I673 of opposing SUN1 KASH-lids. (**g**) Structural details of the SUN1-KASH1 6:6 interface showing interactions between amino acids F671 and I673 of opposing SUN1 KASH-lids that are supported by KASH1 amino acid F8784, but with no interface-spanning interactions between opposing KASH1 chains.

The online version of this article includes the following figure supplement(s) for figure 3:

**Figure supplement 1.** SUN-KASH crystal structures.

**Figure supplement 2.** SUN1-KASH4 forms 6:6 and 12:12 complexes upon sequestration of bound zinc SEC-MALS analysis of SUN1-KASH4 (yellow) and following the removal of bound zinc (demonstrated in *Figure 3e*) by treatment with EDTA (blue).

and providing analogous interface-spanning interactions (*Figure 3a,b*). However, an important distinction is that a torsional rotation of approximately 20° between the 3:3 complexes of SUN1-KASH5, relative to SUN1-KASH4, brings together opposing KASH-lids and enables their interaction across the interface (*Figure 3f*). Thus, tip-to-tip interactions via amino acids I673 and F671 of opposing SUN1 KASH-lids contribute to the extensive 6:6 interface of SUN1-KASH5 (*Figure 3f*).

The SUN1-KASH1 6:6 complex is formed solely of a tripod of KASH-lid tip-to-tip interactions mediated by amino acids I673 and F671, in the same manner and owing to the same torsional rotation as in the SUN1-KASH5 structure (*Figure 3g* and *Figure 3—figure supplement 1c*). KASH1 undergoes acute angulation away from the 6:6 interface (*Figure 3a,b*), as previously observed in SUN2-KASH1/2 (*Figure 1—figure supplement 1e*). As such, whilst amino acid F8784 binds to the KASH-lids of each tip-to-tip interaction site (*Figure 3g*), KASH1's N-terminus does not contribute to the 6:6 interface (*Figure 3a,b*). This creates an open interface, with large solvent channels between opposing 3:3 complexes (*Figure 3—figure supplement 1c*). Overall, the three structures demonstrate alternative SUN-KASH 6:6 interaction mechanisms that are differentially exploited by KASH proteins.

Our findings of differential 6:6 assembly mechanisms raise the possibility that the same SUN1-KASH 6:6 complex could be supported by distinct interfaces. We confirmed this hypothesis for SUN1-KASH4 through the finding that the 6:6 complex is retained upon zinc removal by pre-incubation with EDTA (*Figure 3e* and *Figure 3—figure supplement 2*), likely through reversal to a KASH1-like interface in which the head-to-head interaction is mediated solely by SUN1 amino acids. The zinc-stripped SUN1-KASH4 complex also formed a prominent 12:12 species (*Figure 3—figure supplement 2*), suggesting that in absence of metal coordination, KASH4 can mediate interactions between KASH1-like 6:6 complexes, which could occur through disulphide bond formation of exposed C381 and C382 amino acids. These findings illustrate how SUN1-KASH4 and SUN1-KASH1 represent either ends of a spectrum of possible inter-trimer interfaces in which 6:6 structures are supported solely by KASH-mediated metal coordination and SUN1's KASH-lids, respectively. In contrast, SUN1-KASH5 is an intermediate structure that utilises both KASH and KASH-lid mechanisms to form a fully enclosed 6:6 interface.

## SUN1-KASH1 complex formation depends on KASH-lid 6:6 interactions

On the basis of our SUN-KASH crystal structures, we predicted that KASH-lid tip-to-tip interactions are essential for 6:6 hetero-oligomer formation in solution by SUN1-KASH1 but not SUN1-KASH4. We tested this by generating glutamate mutations of KASH-lid tip amino acids I673 and F671, which mediate interface-spanning tip-to-tip interactions within SUN1-KASH1 and SUN1-KASH5 but have no contacts within their respective 3:3 complexes (*Figure 3c,f,g* and *Figure 4—figure supplement 1a–c*). We also analysed a glutamate mutation of amino acid W676, which mediates hydrophobic interactions with the KASH domain within a constituent 3:3 complex (*Figure 4—figure supplement 1a–c*), and acted as a negative control in disrupting all three SUN-KASH complexes (*Figure 4a,b*). It was not possible to analyse SUN-KASH binding through amylose pull-down owing to the non-specific binding between SUN1 and amylose resin (*Figure 4—figure supplement 1e*). Instead, we exploited this phenomenon by using amylose resin to purify complexes and dissociated proteins following GCN4-SUN1 and MBP-KASH co-expression, which we enriched by ion exchange (*Figure 4—*

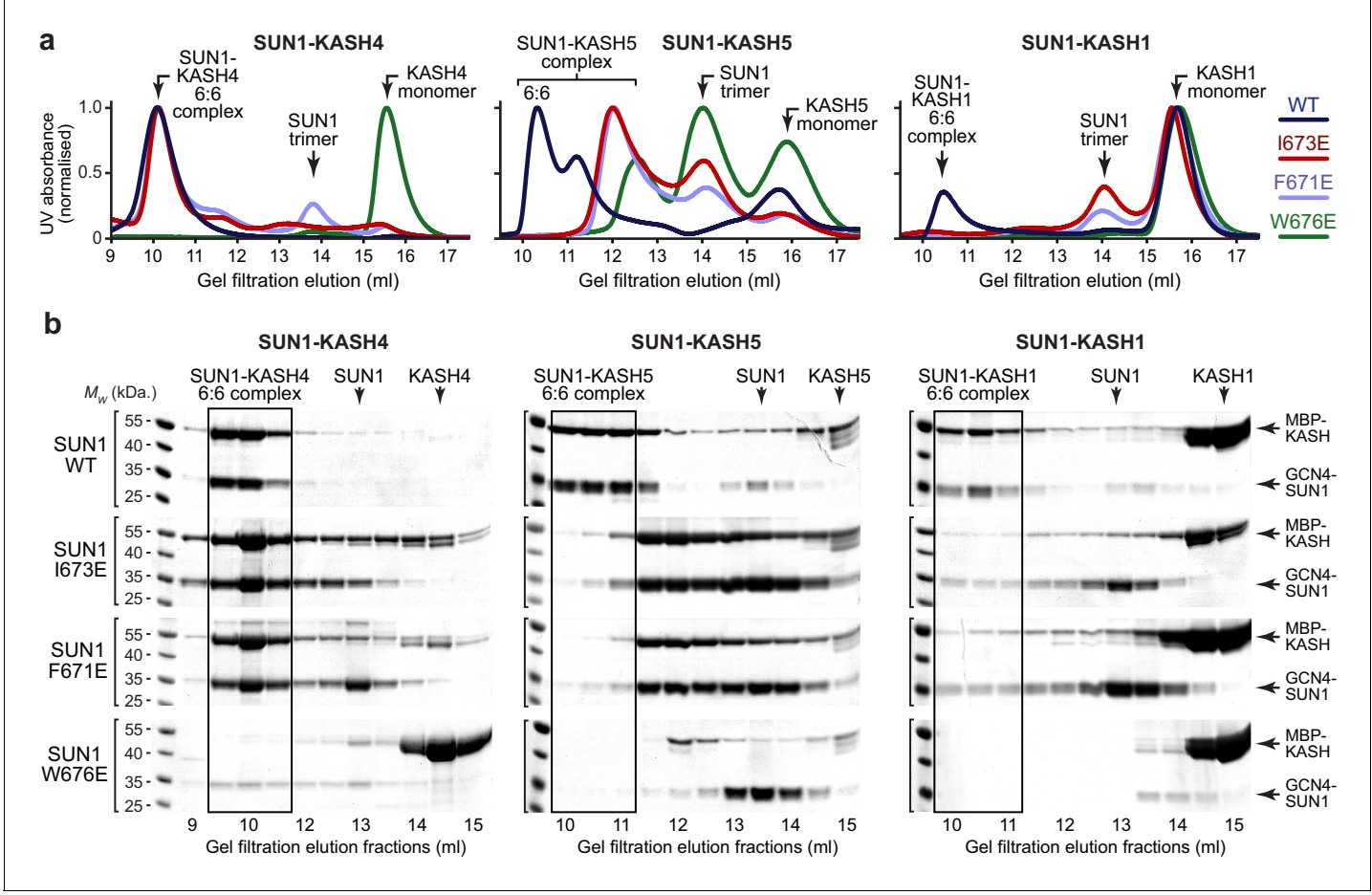

**Figure 4.** SUN1 KASH-lid residues involved in 6:6 assembly are essential for KASH1-binding. (a,b) Gel filtration analysis. GCN4-SUN1 and MBP-KASH proteins were co-expressed and purified by amylose affinity (utilising non-specific binding by SUN1 for non-interacting mutants) and ion exchange (*Figure 4—figure supplement 1d*), and all fractions containing SUN-KASH complexes and dissociated proteins were concentrated and loaded onto an analytical gel filtration column. The elution profiles were validating by SEC-MALS in which wild-type fusion complexes and dissociated GCN4-SUN1 and MBP-KASH1 proteins were found to be 6:6 complexes, trimers and monomers, respectively (*Figure 4—figure supplement 1f*). (a) Gel filtration chromatograms (UV absorbance at 280 nm) across elution profiles for SUN1 wild-type (WT; dark blue), I673E (red), F671E (light blue), and W676E (green), with KASH4 (left), KASH5 (middle), and KASH1 (right), and (b) SDS-PAGE of their corresponding elution fractions. Representative of three replicates using different protein preparations. Source data are provided in *Figure 4—source data 1*.

The online version of this article includes the following source data and figure supplement(s) for figure 4:

**Source data 1.** Uncropped gel images relating to *Figure 4b*.
**Figure supplement 1.** SUN-KASH complex formation upon SUN1 KASH-lid mutagenesis.
**Figure supplement 2.** Biophysical analysis of the SUN1 I673E mutant.

figure supplement 1d), and then pooled all fractions containing SUN-KASH complexes and dissociated proteins for analysis by analytical gel filtration (*Figure 4a,b*). We validated the resulting elution profiles through SEC-MALS by confirming that the wild-type fusion complexes and dissociated GCN4-SUN1 and MBP-KASH proteins are 6:6 complexes, trimers and monomers, respectively (*Figure 4—figure supplement 1f*).

The SUN1-KASH4 6:6 complex was impervious to KASH-lid mutations I673E and F671E (*Figure 4a,b*, *Figure 4—figure supplement 1d* and *Figure 4—figure supplement 2a*), in keeping with the lack of KASH-lid tip-to-tip interactions at its 6:6 interface and the aforementioned reversal to KASH1-like binding only upon stripping of bound zinc In stark contrast, SUN1-KASH1 was disrupted by I673E and F671E mutations (*Figure 4a,b* and *Figure 4—figure supplement 1d*), confirming that KASH-lid tip-to-tip interactions are essential for its 6:6 complex formation. Upon removal of the trimerising GCN4 tag, the dissociated SUN1 I673E protein was monomeric, matching our

observations for wild-type SUN1, which remains monomeric in absence of KASH-binding (*Figure 4—figure supplement 2b*). Further, SAXS analysis confirmed that its SUN domain remained folded (*Figure 4—figure supplement 2c–g* and *Table 2*). The failure to observe smaller hetero-oligomers demonstrates that SUN1-KASH1 3:3 complexes are unstable in absence of the 6:6 interface, indicating that SUN1-KASH1 is a constitutive 6:6 hetero-oligomer.

In agreement with the equal roles of KASH domain and KASH-lid interactions at its 6:6 interface, SUN1-KASH5 exhibited intermediate phenotypes upon I673E and F671E mutation, with retention of complex formation but reduction in oligomer size to species that likely reflect partially dissociating 6:6 complexes (*Figure 4a,b* and *Figure 4—figure supplement 1d*). We conclude that the diverse roles of KASH-lids at the 6:6 interfaces of SUN1-KASH crystal structures are truly reflective of their solution states and that KASH-lid tip-to-tip interactions are essential for assembly of a constitutive SUN1-KASH1 6:6 hetero-oligomer.

## SUN2-KASH complexes form 6:6 and higher molecular weight structures

LINC complexes are commonly formed of SUN1 and SUN2 (*Lei et al., 2009*; *Zhang et al., 2009*), raising the question of whether SUN2 forms similar 6:6 complexes or distinct LINC complex structures? To address this, we purified SUN2 complexes with the three characteristic KASH proteins. SUN2-KASH4 was stable during purification (*Figure 5a*) and SEC-MALS analysis confirmed that it constitutes a 6:6 hetero-oligomer (*Figure 5b*). In contrast, SUN2-KASH5 and SUN2-KASH1 proved to be less stable and more heterogeneous than their comparative SUN1 complexes (*Figure 5a*), and underwent substantial dissociation to SUN2 trimers and KASH monomers during SEC-MALS analysis (*Figure 5c,d*). Nevertheless, eluted SUN2-KASH5 and SUN1-KASH1 complexes are molecular

**Table 2.** Summary of SEC-SAXS data.

| | SUN1 I673E (monomer) | Sun1-kash4 (6:6) | Sun1-kash5 (6:6) | Sun1-kash1 (6:6) |
|---|---|---|---|---|
| SASDBD accession | SASDJF5 | SASDJC5 | SASDJD5 | SASDJE5 |
| Guinier analysis | | | | |
| $I(0)$ (cm$^{-1}$) | 0.042 | 0.045 | 0.100 | 0.130 |
| $Rg$ (Å) | 21 | 40 | 38 | 39 |
| $q_{min}$ (Å$^{-1}$) | 0.0080 | 0.0014 | 0.0070 | 0.0090 |
| $P(r)$ analysis | | | | |
| $I(0)$ (cm$^{-1}$) | 0.042 | 0.045 | 0.102 | 0.132 |
| $Rg$ (Å) | 22 | 40 | 39 | 39 |
| $D_{max}$ (Å) | 82 | 135 | 135 | 130 |
| Porod volume (Å$^3$) | 39,367 | 292,301 | 274,824 | 303,602 |
| MW from Porod volume (kDa) | 23 | 172 | 162 | 179 |
| $V_C$ (Å$^2$) | 238 | 825 | 784 | 853 |
| MW from $V_C$ (kDa) | 22 | 139 | 131 | 152 |
| *DAMMIF ab initio modelling (30 models)* | | | | |
| Symmetry | P1 | N/A | N/A | N/A |
| *NSD* mean | 0.645 | N/A | N/A | N/A |
| $\chi^2$ (reference model) | 1.85 | N/A | N/A | N/A |
| *Structural modelling* | | | | |
| CRYSOL - crystal structure ($\chi^2$) | 5.43 | 1.62 | 5.50 | 4.83 |
| CORAL - modelling of N-termini ($\chi^2$) | N/A | 1.25 | 1.70 | 4.55 |
| CORAL - rigid body modelling ($\chi^2$) | N/A | N/A | N/A | 1.56 |
| SREFLEX - normal mode analysis ($\chi^2$) | 1.72–1.98 | N/A | N/A | N/A |

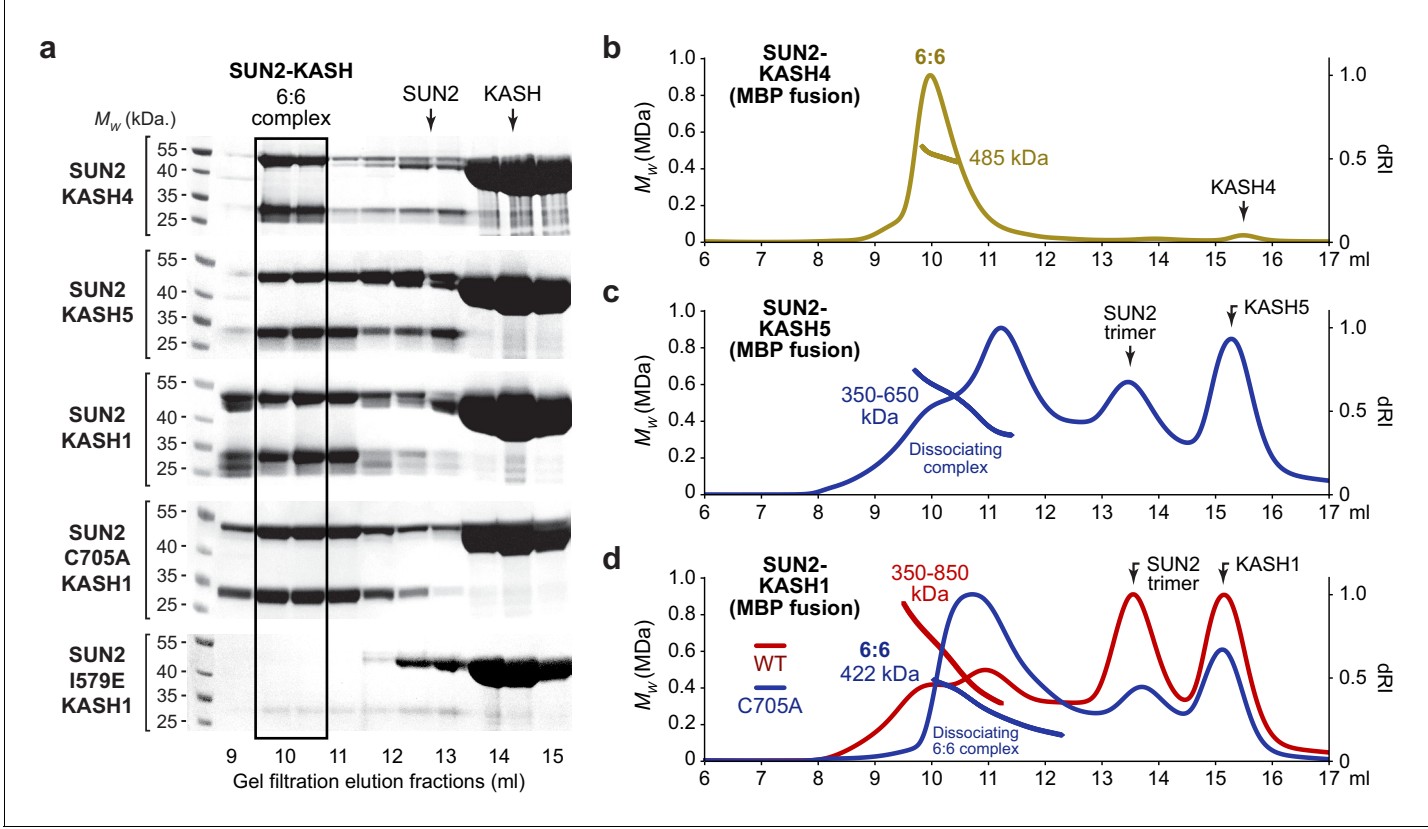

**Figure 5.** SUN2 forms 6:6 and higher molecular weight SUN-KASH complexes through head-to-head assembly. (**a**) Gel filtration analysis shown as SDS-PAGE of elution fractions. GCN4-SUN2 (wild-type, C705A and I579E) and MBP-KASH proteins were co-expressed and purified by amylose affinity (utilising non-specific binding by SUN2 for non-interacting mutants) and ion exchange, and all fractions containing SUN-KASH complexes and dissociated proteins were concentrated and loaded onto an analytical gel filtration column. Source data are provided in *Figure 5—source data 1*. (**b**–**d**) SEC-MALS analysis of SUN2-KASH (MBP fusion) complexes following gel filtration elution (**a**). (**b**) SUN2-KASH4 is a 6:6 complex of 485 kDa (theoretical – 463 kDa). (**c**) SUN2-KASH5 forms a range of molecular species of at least 350–650 kDa, suggesting dissociation across the elution profile of 6:6 and larger complexes (theoretical 3:3 and 6:6–232 kDa and 463 kDa). (**d**) SUN2-KASH1 wild-type (red) forms a range of molecular species of at least 350–850 kDa, whilst the SUN2 C705A mutation (blue) stabilises a 6:6 complex of 422 kDa, suggesting dissociation across the elution profile of 6:6 and larger complexes (theoretical 3:3 and 6:6–232 kDa and 465 kDa).

The online version of this article includes the following source data for figure 5:

**Source data 1.** Uncropped gel images relating to *Figure 5a*.

species of 350–650 kDa and 350–850 kDa, respectively (*Figure 5c,d*), which are substantially larger than 3:3 complexes (232 kDa) and include 6:6 complexes (463 and 465 kDa). Thus, SEC-MALS profiles likely represent dissociation from SUN2-KASH complexes of 6:6 and higher order hetero-oligomers. We confirmed this for SUN2-KASH1 by introducing SUN2 mutation C705A (designed to prevent disulphide bond formation and hence minimise heterogeneity), which removed higher order structures and demonstrated the presence of dissociating 6:6 hetero-oligomers (*Figure 5d*). Finally, we introduced SUN2 mutation I579E, which targets the inter-trimer interface in precisely the same manner as SUN1 mutation I673E. The SUN2 mutation I579E fully disrupted the SUN2-KASH1 complex, mimicking the phenotype of SUN1 I673E mutation in SUN1-KASH1, confirming that KASH-lid tip-to-tip interactions are essential for assembly of SUN2-KASH1 complexes. Thus, we conclude that despite their lower stability and greater heterogeneity, SUN2-KASH complexes are 6:6 and higher order structures, and interactions that solely span the 6:6 interface are essential for SUN1-KASH1 complex formation.

## Hinge-like motion of the SUN-KASH 6:6 interface

How could the SUN-KASH 6:6 complex be orientated within the nuclear envelope? Its head-to-head assembly suggests a horizontal orientation, parallel to the outer nuclear membrane, with SUN trimers organised obliquely within the peri-nuclear space. In this configuration, tension forces carried by SUN and KASH molecules would exert bending moments on the structure, favouring a hinge-like angulation between opposing 3:3 complexes. We thus utilised small-angle X-ray scattering (SAXS) to determine whether SUN-KASH complexes adopt angled conformations in solution. Whilst SAXS data of SUN1-KASH4 and SUN1-KASH5 were closely fitted by their crystal structures upon flexible modelling of missing termini ($\chi^2$ values of 1.25 and 1.70), we achieved only poor fits for SUN1-KASH1 ($\chi^2$ = 4.83) (*Figure 6a,b*, *Figure 6—figure supplement 1* and *Table 2*). In case of large-scale motion, we performed SAXS-based rigid-body modelling using two SUN1-KASH1 3:3 complexes as independent rigid bodies. We consistently obtained models that closely fitted experimental data ($\chi^2$ = 1.56) in which 3:3 complexes interact head-to-head with a bend of approximately 60° relative to the crystal structure (*Figure 6c,d* and *Table 2*). In this model, two pairs of KASH-lid tip-to-tip interactions by I673 and F671 are retained, whilst the third is disrupted, and an additional interface is formed between opposing central KASH-lids. Thus, KASH-lids may act as a hinge at the 6:6 interface, allowing the linear crystal structure to open into a continuous range of angled conformations, including (but not limited to) the 60° angulation predicted by SAXS analysis.

The hinged SUN1-KASH1 structure solves a critical problem in understanding the potential role of the 6:6 complex within its cellular context. Whilst the linear crystal structure distributes the KASH1 N-termini around its circumferential exterior (*Figures 1c* and *3b*), making it difficult to envisage how all KASH1 molecules could access the outer nuclear membrane, the asymmetrical hinged structure places all six KASH1 N-termini in favourable positions and orientations for their upstream transmembrane sequences to cross the outer nuclear membrane (*Figure 6c,d*).

Is a similar hinge-like angulation possible for SUN1-KASH4 and SUN1-KASH5? Whilst their extensive 6:6 interfaces retain linear structures in solution (*Figure 6a–b*, *Figure 6—figure supplement 1* and *Table 2*), angulation may be achieved by tension forces. We thus performed normal mode analysis to determine whether angled structures are conformationally accessible. We observed low-frequency normal modes corresponding to hinge-like angulation at the 6:6 interface for all SUN-KASH complexes (*Figure 7*), indicating that angled conformations are accessible flexible states. As described for SUN1-KASH1, hinging of SUN1-KASH4 and SUN-KASH5 would place the N-termini of their constituent KASH domains in suitable positions and orientations to cross the outer nuclear membrane, so adoption of hinged conformations may be a critical part of forming stable membrane-associated assemblies. We thus conclude a model in which hinged SUN-KASH 6:6 complexes, parallel with the outer nuclear membrane, act as nodes for the integration and distribution of tension forces between oblique SUN trimers and KASH molecules within a branched LINC complex network (*Figure 8*).

## Discussion

How does our finding of a constitutive SUN-KASH 6:6 assembly integrate with previous biochemical studies of the LINC complex? It was previously shown by analytical ultracentrifugation, SEC-MALS, and gel filtration that luminal SUN2 is trimeric, and its isolated SUN domain is a trimer or monomer, depending on biochemical conditions (*Zhou et al., 2012*; *Wang et al., 2012*; *Sosa et al., 2013*; *Jahed et al., 2018b*). These findings agree with our observations that the isolated SUN domain of SUN1 becomes monomeric upon cleavage of its N-terminal GCN4 expression tag (which mimics the trimeric luminal coiled-coil), so is entirely dependent on KASH-binding to stabilise its trimeric structure and head-to-head assembly. The only previous analysis of SUN-KASH in solution involved demonstrating complex formation by analytical gel filtration, without means for oligomer determination (*Esra Demircioglu et al., 2016*). Thus, the 3:3 SUN-KASH model was the natural conclusion of combining SUN's luminal trimer with the extensive 3:3 complexes within SUN2-KASH1/2 crystal lattices (*Sosa et al., 2012*; *Wang et al., 2012*). Our SEC-MALS and SEC-SAXS analyses provide the first reported evidence of solution structure, revealing that SUN-KASH complexes formed by SUN1 and SUN2 are 6:6 hetero-oligomers in which 3:3 structures are locked in head-to-head interactions, as observed in our SUN1-KASH crystal structures and in previous SUN2-KASH crystal lattices (*Sosa et al., 2012*; *Wang et al., 2012*). Further, mutational analysis confirmed that SUN1/2-KASH1

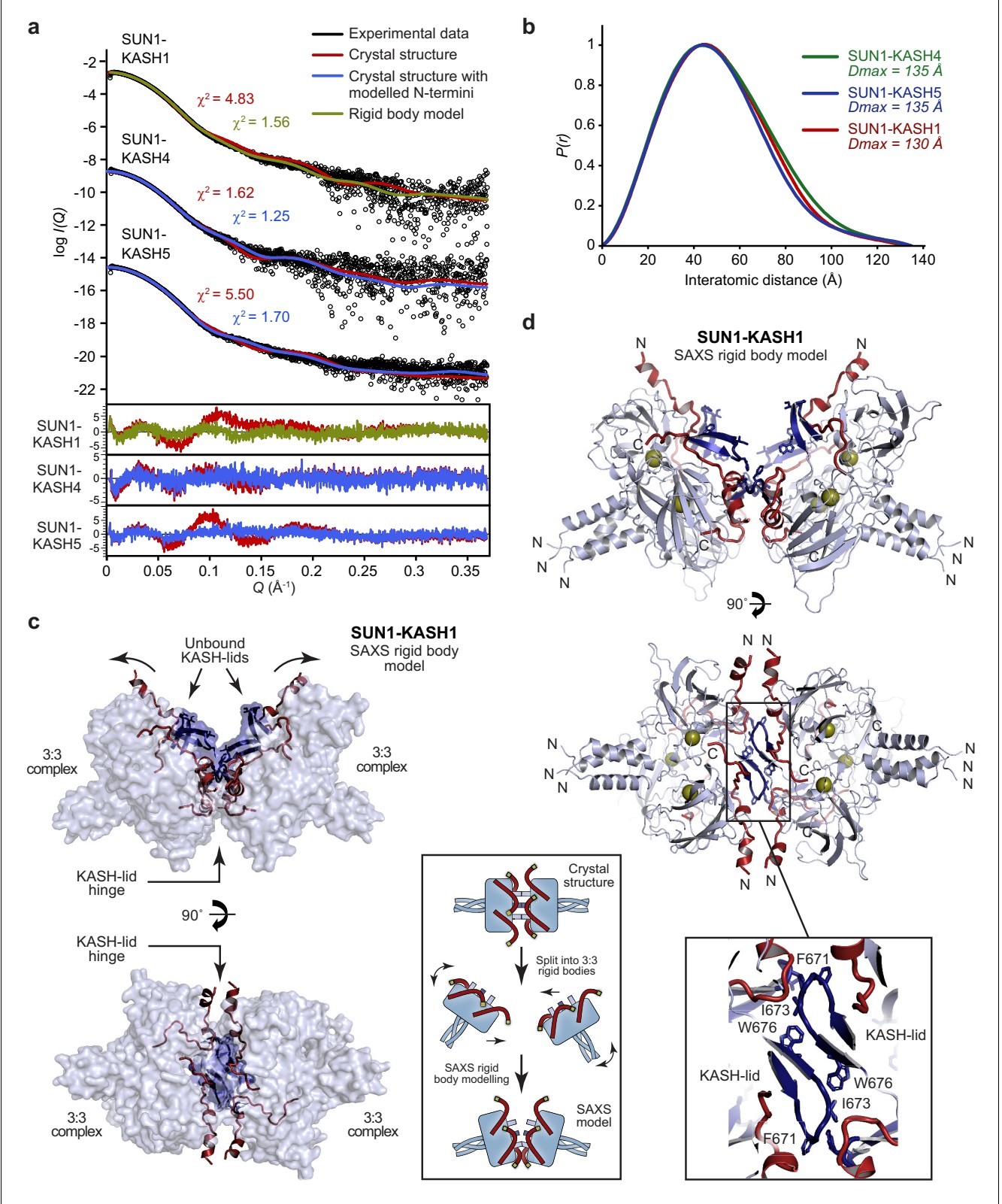

**Figure 6.** SEC-SAXS analysis of SUN-KASH 6:6 complexes. (**a**) SAXS scattering curves of SUN1-KASH4, SUN1-KASH5, and SUN1-KASH1 overlaid with theoretical scattering curves of their crystal structures (red), crystal structures with KASH flexible N-termini modelled by CORAL (blue) and rigid body model of two 3:3 complexes (green). Residuals for each fit are shown (inset). Representative of more than three replicates using different protein preparations. (**b**) SAXS *P(r)* distributions showing maximum dimensions of 135 Å, 135 Å, and 130 Å, respectively. (**c–d**) SAXS rigid body model of SUN1-

*Figure 6 continued on next page*

*Figure 6 continued*

KASH1 shown as (**c**) surface and (**d**) cartoon representation, in which two constituent 3:3 complexes from its crystal structure were assigned as rigid bodies, with the 6:6 assembly generated by fitting to experimental SAXS data of solution SUN1-KASH1 ($\chi^2$ = 1.56). The inlet schematic illustrates the SAXS rigid body modelling procedure in which the crystal structure was split into its constituent 3:3 complexes, which were rotated as rigid bodies in three dimensions and allowed to interact, whilst fitted against experimental SAXS data. (**d**) The cartoon representation highlights structural details of the predicted KASH-lid interface, including the presence of unbound KASH-lids, and the close approximation of opposing KASH-lids, which achieve an asymmetric positioning of the N-termini of KASH domains in locations and orientations compatible with their upstream sequences crossing the outer nuclear membrane.

The online version of this article includes the following figure supplement(s) for figure 6:

**Figure supplement 1.** SAXS analysis of SUN-KASH complexes.

complexes depend on interactions across the 6:6 interface for their stability. Hence, our conclusion that SUN-KASH complexes are 6:6 hetero-oligomers in vitro is consistent with all existing crystallographic, biochemical, and biophysical data.

How does the SUN-KASH 6:6 assembly relate to previous observations of LINC complex structure and function within the cell? The oligomeric states of luminal regions of SUN1 and SUN2, upon expression and targeting to the nuclear envelope, were determined by fluorescence fluctuation spectroscopy as trimers with additional higher order SUN1 structures (*Hennen et al., 2017*; *Hennen et al., 2018*). In these studies, expressed KASH domains and isolated SUN domains remained mostly monomeric, suggesting that expressed constructs did not form SUN-KASH complexes with endogenous partners. Hence, these studies provided important evidence that the coiled-coils of SUN's luminal regions form trimers and larger oligomers but did not determine the stoichiometry of SUN-KASH complexes. The assembly of higher order LINC structures has also been suggested by numerous other cellular findings, including immobility within the nuclear envelope (*Lu et al., 2008*), foci formation within the meiotic nuclear envelope (*Ding et al., 2007*; *Morimoto et al., 2012*; *Horn et al., 2013b*), and the formation of transmembrane actin-associated nuclear (TAN) lines (*Luxton et al., 2010*). Our model of LINC complex branching by SUN-KASH 6:6 assembly is consistent with the observed oligomeric state of SUN's luminal region and higher order LINC assembly, but its molecular details are not directly tested by any existing cellular data. Thus, our molecular model of a branched LINC complex, and similarly the role of zinc-binding in the SUN1-KASH4 complex, must be tested in future studies of the consequence of separation of function mutations (such as targeting the 6:6 interfaces of SUN1/2-KASH1 complexes by I673E and I579E mutations) on the cellular structure and function of the LINC complex.

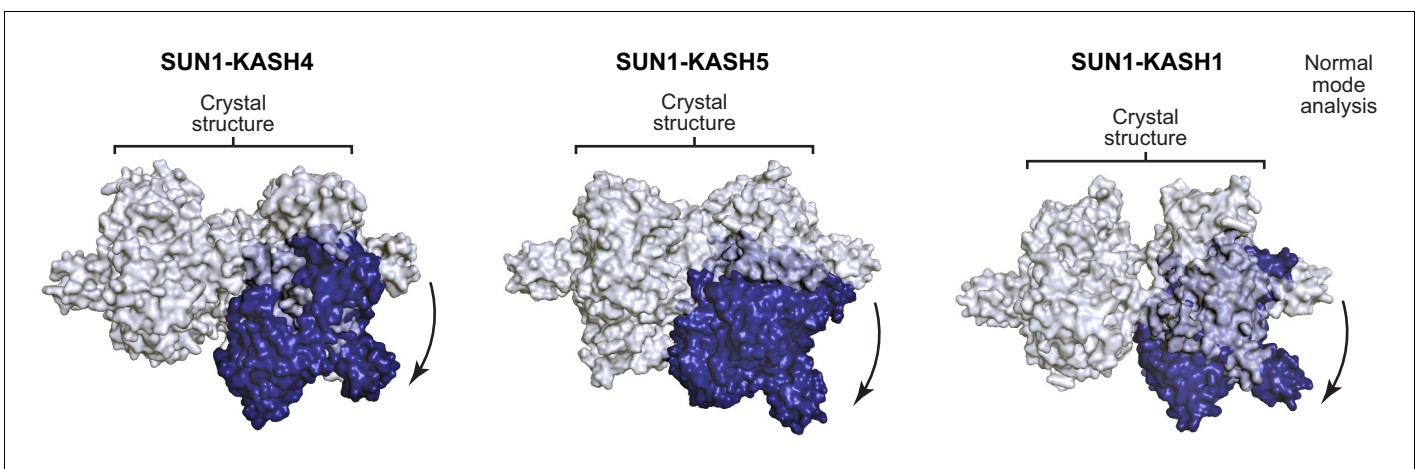

**Figure 7.** Hinge-link conformational flexibility within SUN-KASH 6:6 assemblies. Normal mode analysis of SUN-KASH complexes in which non-linear normal modes calculated by the NOLB algorithm are shown as the largest amplitude of motion of one constituent 3:3 complex (blue) relative to its original position and its stationary opposing 3:3 complex within the crystal structure (grey) for SUN1-KASH4 (left), SUN1-KASH5 (middle), and SUN1-KASH1 (right).

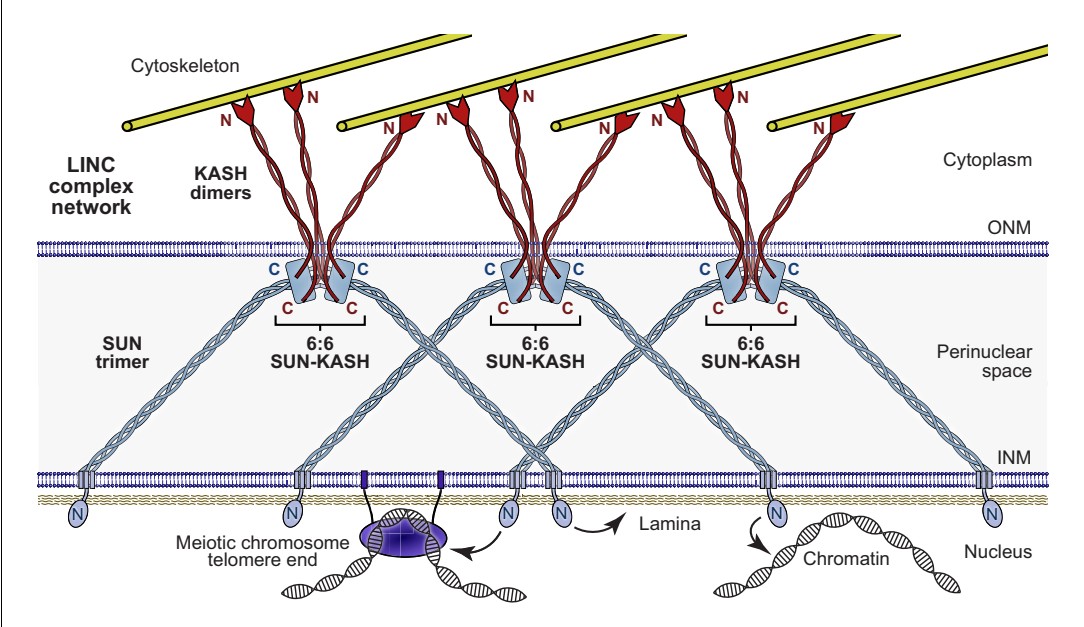

**Figure 8.** The Linker of Nucleoskeleton and Cytoskeleton (LINC) complex as a branched network of SUN-KASH assemblies. Model of the LINC complex as a branched network in which SUN-KASH 6:6 complexes act as nodes for force integration and distribution between two SUN trimers and three KASH dimers, which can bind to spatially separated and distinct nuclear and cytoskeletal components, respectively. This model enables cooperation between adjacent molecules within a LINC complex network to facilitate the transduction of large and coordinated forces across the nuclear envelope.

The advantages of a branched LINC complex network include its ability to transmit large forces, being impervious to the breakage of individual linkages, and in mediating communication and coordination between adjacent molecules. The SUN-KASH 6:6 assembly provides an attractive structural means for branching, which may combine with a series of episodic instances of oligomer variation along the SUN-KASH axis to generate a highly branched LINC complex network. Firstly, oligomer variation could occur through higher order assembly of SUN-KASH 6:6 complexes, as indicated by our observation of higher order SUN2-KASH structures and the formation of SUN1-KASH4 12:12 complexes upon disruption of zinc-binding. Secondly, oligomeric variation within SUN's luminal regions may mediate branching, such as indicated by the formation of trimers and larger oligomer by luminal SUN1 (*Hennen et al., 2018*) and disulphide bond formation by SUN1 amino acid C526 (*Lu et al., 2008*). Finally, oligomer variation between SUN and KASH proteins could mediate branching across the outer nuclear membrane. Indeed, KASH5 is dimeric (Gurusaran and Davies, unpublished findings), raising the question of how SUN trimers and KASH dimers are organised into discrete 6:6 complexes? We suggest that each KASH dimer likely spans both SUN trimers, thereby establishing a symmetrical array of SUN1-KASH interfaces within each 6:6 structure, which constitute branching events between SUN-KASH5 6:6 complexes and their dimeric cytoskeletal attachments. Thus, we propose that coordinated force transduction is achieved by a highly branched LINC complex network in which SUN-KASH 6:6 hetero-oligomers contribute to branching by mediating force distribution and integration between three KASH dimers and two SUN trimers (*Figure 8*).

The head-to-head nature of SUN-KASH 6:6 complexes suggests their orientation parallel to the outer nuclear membrane, with SUN trimers organised obliquely within the peri-nuclear space (*Figure 8*). Our SAXS analysis of SUN1-KASH1 indicated that it adopts a hinged conformation in solution, stabilised by two KASH-lid tip-to-tip interactions and laterally associated central KASH-lids. Whilst hinged motions were not required to explain SAXS data of SUN1-KASH4/5, normal mode analysis predicted that hinged structures of up to approximately 60° angulation are conformationally accessible states for all three SUN1-KASH complexes. Thus, we suggest that all SUN-KASH head-to-head structures can undergo hinge-like motion at their 6:6 interface, with a large proportion of highly angled conformations accounting for their dominance in the SAXS data of SUN1-KASH1 but

not other complexes. This hinge-like motion would result in SUN-KASH complexes becoming angled in response to the magnitude and direction of tension forces carried by SUN and KASH molecules, whilst providing the conformational flexibility necessary for constituent KASH proteins to adopt orientations that allow upstream transmembrane sequences to cross the outer nuclear membrane. Further, all three SUN-KASH 6:6 interfaces are largely hydrophobic, so could be stabilised by interactions with phospholipids, possibly as part of integrated membrane-bound complexes that include KASH's transmembrane regions. Thus, hinge-like flexibility of SUN-KASH may result in a diverse range of angled conformations owing to distinct tension forces, steric constraints and membrane structures of particular spatiotemporal environments.

What are the roles of distinct SUN and KASH proteins in LINC complex structure and function? Whilst SUN1 and SUN2 form similar 6:6 hetero-oligomers, we observed notable differences in the stability and higher assembly of their SUN-KASH complexes. The reduced stability of SUN2-KASH complexes could facilitate a faster turnover of SUN2-containing LINC complexes, whilst higher order assembly of SUN2-KASH may combine with differential SUN1/2 luminal assemblies (*Hennen et al., 2017*; *Hennen et al., 2018*) to achieve distinct LINC complex architectures. These findings may underlie some of the observed asymmetries between SUN1 and SUN2 LINC complexes, such as their differential preference for cytoskeletal components and their non-redundant functions (*Link et al., 2014*; *Zhu et al., 2017*; *Thakar et al., 2017*; *May and Carroll, 2018*). It is important to note that SUN2-KASH4, in which the 6:6 interface is mediated solely by KASH4 zinc sites, is the only SUN2 complex that retains the high affinity observed for SUN1 complexes. In contrast, SUN amino acids contribute to the 6:6 interfaces of KASH1/5 complexes, explaining how SUN protein sequence diversity can account for the substantially reduced affinity of SUN2-KASH1/5 in comparison with their SUN1 complexes. The variation of KASH proteins seemingly provides even greater functional diversity given their entirely non-redundant roles. An intriguing observation is that Nesprin-4 and KASH5, which transduce microtubule forces (*Horn et al., 2013a*; *Roux et al., 2009*; *Morimoto et al., 2012*; *Horn et al., 2013b*), demonstrate extensive interactions at their 6:6 interfaces. In contrast, a far less extensive 6:6 interface is found in classical Nesprins, which transduce actin forces and the tensile strength of intermediate filaments (*Banerjee et al., 2014*; *Starr and Fridolfsson, 2010*; *Ketema and Sonnenberg, 2011*). Thus, cytoskeletal components may have differential requirements for the strength, structure and stability of SUN-KASH 6:6 hetero-oligomers. Further, differences in regulatory mechanisms, such as zinc-binding in SUN1-KASH4 assembly, may contribute towards specialisation. The expression levels and relative availability of SUN and KASH proteins will determine their incorporation into LINC complexes, and specialised functionalities may be achieved by combining distinct isoforms within the same LINC complex network or within separate networks of the same cell.

How is LINC complex assembly regulated within the cell? An intriguing finding is that SUN proteins undergo autoinhibition, in which SUN domains become bound by upstream sequences in monomeric conformations that are incapable of binding to KASH domains (*Nie et al., 2016*; *Xu et al., 2018*; *Jahed et al., 2018b*; *Jahed et al., 2018a*). These autoinhibitory conformations likely represent unassembled states that may be crucial intermediates in the dynamic process of LINC complex expression, localisation, and assembly within the cell. They may also represent a 'storage form' of SUN proteins that form when quantities of available KASH proteins are limiting. This would establish discrete pools of assembled and unassembled SUN proteins, which could play an important role in preventing unbound SUN molecules from weakening established LINC structures by continually competing for KASH-binding. Further, given the myriad of LINC complex functions in almost all eukaryotic cells (*Lee and Burke, 2018*; *Meinke and Schirmer, 2015*; *Starr and Fridolfsson, 2010*), assembly is likely directed along specific pathways to achieve distinct LINC complex structures for the fulfilment of specialised functions. Thus, regulatory processes must overcome autoinhibition, enable KASH-binding, and direct LINC assembly in a timely manner. These may involve chaperones, enzymatic modification, protein interactions, and/or chemical conditions of the nuclear envelope environment. In specific, these may include regulation by luminal ion concentration and pH (*Jahed et al., 2018b*), local regulation of SUN-KASH angulation, control of SUN1-KASH4 assembly by zinc availability, and determining the nature of LINC complexes through relative availability of SUN and KASH protein isoforms. We have hitherto considered variations within SUN-KASH 6:6 complexes, but also recognise the potential for regulatory mechanisms of the nuclear envelope to induce more substantial structural changes. Thus, whilst our model of LINC complex branching through

SUN-KASH 6:6 assembly is consistent with all existing data, it remains possible that alternative LINC complex conformations may form within the spatial and temporal contexts of disparate cell types.

# Materials and methods

## Key resources table

| Reagent type (species) or resource | Designation | Source or reference | Identifiers | Additional information |
|---|---|---|---|---|
| Gene (*Homo sapiens*) | SUN1 | GeneArt | O94901 | |
| Gene (*Homo sapiens*) | SUN2 | GeneArt | Q9UH99 | |
| Gene (*Homo sapiens*) | Nesprin-1 | GeneArt | Q8NF91 | |
| Gene (*Homo sapiens*) | Nesprin-4 | GeneArt | Q8N205 | |
| Gene (*Homo sapiens*) | KASH4 | GeneArt | Q8N6L0 | |
| Recombinant DNA reagent | pRSF-Duet1-SUN1 (plasmid) | This paper | | SUN1 (616–812) cloned into a pRSF-Duet1 vector |
| Recombinant DNA reagent | pRSF-Duet1-SUN1 I673E (plasmid) | This paper | | SUN1 (616–812) I673E cloned into a pRSF-Duet1 vector |
| Recombinant DNA reagent | pRSF-Duet1-SUN1 F671E (plasmid) | This paper | | SUN1 (616–812) F671E cloned into a pRSF-Duet1 vector |
| Recombinant DNA reagent | pRSF-Duet1-SUN1 W676E (plasmid) | This paper | | SUN1 (616–812) W676E cloned into a pRSF-Duet1 vector |
| Recombinant DNA reagent | pRSF-Duet1-SUN2 (plasmid) | This paper | | SUN2 (522–717) cloned into a pRSF-Duet1 vector |
| Recombinant DNA reagent | pMAT11-KASH1 (plasmid) | This paper | | Nesprin-1 (8769–8797) cloned into a pMAT11 vector |
| Recombinant DNA reagent | pMAT11-KASH4 (plasmid) | This paper | | Nesprin-4 (376–404) cloned into a pMAT11 vector |
| Recombinant DNA reagent | pMAT11-KASH5 (plasmid) | This paper | | KASH5 (542–562) cloned into a pMAT11 vector |
| Strain, strain background (*Escherichia coli*) | Rosetta2 (DE3) | Thermo Fisher | EC0114 | Chemically competent cells |
| Software, algorithm | XDS | http://xds.mpimf-heidelberg.mpg.de/ | | |

*Continued on next page*

*Continued*

| Reagent type (species) or resource | Designation | Source or reference | Identifiers | Additional information |
|---|---|---|---|---|
| Software, algorithm | XSCALE | http://xds.mpimf-heidelberg.mpg.de/html_doc/xscale_program.html | | |
| Software, algorithm | Phaser | PHENIX | | |
| Software, algorithm | PHENIX Autobuild | PHENIX | | |
| Software, algorithm | PHENIX refine | PHENIX | | |
| Software, algorithm | AutoPROC | Global phasing | | |
| Software, algorithm | ASTRA 6 | Wyatt Technology | | |
| Software, algorithm | ScÅtter 3.0 | http://www.bioisis.net | | |
| Software, algorithm | PRIMUS | Atsas | | |
| Software, algorithm | DAMMIF | Atsas | | |
| Software, algorithm | CRYSOL | Atsas | | |
| Software, algorithm | SREFLEX | Atsas | | |
| Software, algorithm | CORAL | Atsas | | |
| Software, algorithm | SAMSON element | https://www.samson-connect.net | | |

## Recombinant protein expression and purification

The SUN domains of human SUN1 (amino acid residues 616–812) and SUN2 (amino acid residues 522–717) were fused to N-terminal TEV-cleavable His$_6$-GCN4 tags (as described in *Sosa et al., 2012*) and cloned into pRSF-Duet1 (Merck Millipore) vectors. The KASH domains of human KASH5 (amino acid residues 542–562), Nesprin-4 (KASH4, amino acid residues 376–404), and Nesprin-1 (KASH1, amino acid residues 8769–8797) were cloned into pMAT11 (*Peränen et al., 1996*) vectors for expression as TEV-cleavable His$_6$-MBP fusion proteins, respectively. SUN and KASH constructs were co-expressed in BL21 (DE3) cells (Novagen), in 2xYT media, induced with 0.5 mM IPTG for 16 hr at 25°C. Cell disruption was achieved by sonication in 20 mM Tris pH 8.0, 500 mM KCl for SUN1-KASH complexes, 20 mM Tris pH 8.0, 150 mM KCl for SUN2-KASH complexes, and cellular debris removed by centrifugation at 40,000 g. Fusion proteins were purified through consecutive Ni-NTA (Qiagen), amylose (NEB), and HiTrap Q HP (GE Healthcare) ion exchange chromatography. TEV protease was utilised to remove affinity tags and cleaved samples were purified through ion exchange chromatography and size exclusion chromatography (HiLoad 16/600 Superdex 200, GE Healthcare) in 20 mM Tris pH 8.0, 150 mM KCl, 2 mM DTT. Protein samples were concentrated using Microsep Advance Centrifugal Devices 10,000 MWCO centrifugal filter units (PALL) and were stored at −80 °C following flash-freezing in liquid nitrogen. Protein samples were analysed by SDS-PAGE and visualised with Coomassie staining. Concentrations were determined by UV spectroscopy using a Cary 60 UV spectrophotometer (Agilent) with extinction coefficients and molecular weights calculated by ProtParam (http://web.expasy.org/protparam/).

## Crystal structure of SUN1-KASH4 (PDB accession 6R16)

SUN1-KASH4 protein crystals were obtained through vapour diffusion in sitting drops, by mixing 100 nl of protein at 25 mg/ml with 100 nl of crystallisation solution (0.06 M $MgCl_2$; 0.06 M $CaCl_2$; 0.1 M Imidazole pH 6.5; 0.1M MES (acid) pH 6.5; 18% Ethylene glycol; 18% PEG 8K) and equilibrating at 20°C for 4–9 days. Crystals were flash frozen in liquid nitrogen. X-ray diffraction data were collected at 0.9795 Å, 100 K, as 2000 consecutive 0.10° frames of 0.040 s exposure on a Pilatus 6 M-F detector at beamline I04 of the Diamond Light Source synchrotron facility (Oxfordshire, UK). Data were indexed and integrated in XDS (*Kabsch, 2010*), scaled in XSCALE (*Diederichs et al., 2003*) and merged using Aimless (*Evans, 2011*). Crystals belong to orthorhombic spacegroup $P2_12_12_1$ (cell dimensions a = 104.37 Å, b = 117.21 Å, c = 138.42 Å, α = 90°, β = 90°, γ = 90°), with six copies of SUN1 and KASH4 per asymmetric unit. The structure was solved by molecular replacement using Phaser (*McCoy et al., 2007*), with SUN1-KASH1 (this study, PDB accession 6R15) as a search model. The structure was re-built by PHENIX Autobuild (*Adams et al., 2010*) and completed through iterative manual model building in Coot (*Emsley et al., 2010*), with the addition of six potassium ions, three zinc ions and ethylene glycol ligands. The structure was refined using PHENIX refine (*Adams et al., 2010*) with isotropic atomic displacement parameters and TLS parameters, using SUN1-KASH1 as a reference structure. The structure was refined against 2.75 Å data to $R$ and $R_{free}$ values of 0.2190 and 0.2549, respectively, with 98.22% of residues within the favoured regions of the Ramachandran plot (0 outliers), clashscore of 4.89 and overall MolProbity score of 1.26 (*Chen et al., 2010*). The final SUN1-KASH4 model was analysed using the *Online_DPI* webserver (http://cluster. physics.iisc.ernet.in/dpi) to determine a Cruikshank diffraction precision index (DPI) of 0.25 Å (*Kumar et al., 2015*).

## Crystal structure of SUN1-KASH5 (PDB accession 6R2I)

SUN1-KASH5 protein crystals were obtained through vapour diffusion in sitting drops, by mixing 100 nl of protein at 25 mg/ml with 100 nl of crystallisation solution (0.12 M 1,6-Hexanediol; 0.12 M 1-Butanol 1,2-Propanediol (racemic); 0.12 M 2-Propanol; 0.12 M 1,4-Butanediol; 0.12 M 1,3-Propanediol; 0.1 M Imidazole pH 6.5; 0.1 M MES (acid) pH 6.5; 18% Glycerol; 18% PEG 4K) and equilibrating at 20°C for 4–9 days. Crystals were flash frozen in liquid nitrogen. X-ray diffraction data were collected at 0.9282 Å, 100 K, as 2000 consecutive 0.10° frames of 0.050 s exposure on a Pilatus 6 M-F detector at beamline I04-1 of the Diamond Light Source synchrotron facility (Oxfordshire, UK). Data were indexed, integrated, scaled, and merged in AutoPROC using XDS (*Kabsch, 2010*) and Aimless (*Evans, 2011*). Crystals belong to hexagonal spacegroup $P6_322$ (cell dimensions a = 80.16 Å, b = 80.16 Å, c = 177.62 Å, α = 90°, β = 90°, γ = 120°), with one copy of SUN1 and KASH5 per asymmetric unit. The structure was solved by molecular replacement using Phaser (*McCoy et al., 2007*), with SUN1-KASH1 (this study, PDB accession 6R15) as a search model. The structure was re-built by PHENIX Autobuild (*Adams et al., 2010*) and completed through iterative manual model building in Coot (*Emsley et al., 2010*), with the addition of a potassium ion. The structure was refined using PHENIX refine (*Adams et al., 2010*), using anisotropic atomic displacement parameters. The structure was refined against 1.54 Å data to $R$ and $R_{free}$ values of 0.1495 and 0.1683, respectively, with 96.71% of residues within the favoured regions of the Ramachandran plot (0 outliers), clashscore of 6.11 and overall MolProbity score of 1.54 (*Chen et al., 2010*). The final SUN1-KASH5 model was analysed using the *Online_DPI* webserver (http://cluster.physics.iisc.ernet.in/dpi) to determine a Cruikshank diffraction precision index (DPI) of 0.06 Å (*Kumar et al., 2015*).

## Crystal structure of SUN1-KASH1 (PDB accession 6R15)

SUN1-KASH1 protein crystals were obtained through vapour diffusion in sitting drops, by mixing 100 nl of protein at 21 mg/ml with 100 nl of crystallisation solution (0.09 M NaF; 0.09 M NaBr; 0.09 M NaI; 0.1M Sodium HEPES pH 7.5; 0.1 M MOPS (acid) pH 7.5; 18% PEGMME 550; 18% PEG 20K) and equilibrating at 20°C for 4–9 days. Crystals were flash frozen in liquid nitrogen. X-ray diffraction data were collected at 0.9282 Å, 100 K, as 2000 consecutive 0.10° frames of 0.100 s exposure on a Pilatus 6 M-F detector at beamline I04-1 of the Diamond Light Source synchrotron facility (Oxfordshire, UK). Data were indexed, integrated, scaled, and merged in Xia2 (*Winter, 2010*) using XDS (*Kabsch, 2010*), XSCALE (*Diederichs et al., 2003*), and Aimless (*Evans, 2011*). Crystals belong to hexagonal spacegroup $P6_322$ (cell dimensions a = 80.45 Å, b = 80.45 Å, c = 182.55 Å, α = 90°,

β = 90°, γ = 120°), with one copy of SUN1 and KASH1 per asymmetric unit. The structure was solved by molecular replacement using Phaser (*McCoy et al., 2007*), with the SUN domain from SUN2-KASH1 (PDB accession 4DXR; 67% sequence identity) (*Sosa et al., 2012*) as a search model. The structure was re-built by PHENIX Autobuild (*Adams et al., 2010*) and completed through iterative manual model building in Coot (*Emsley et al., 2010*), with the addition of a potassium ion, and PEG and HEPES ligands. The structure was refined using PHENIX refine (*Adams et al., 2010*), using iso-tropic atomic displacement parameters with four TLS groups per chain. The structure was refined against 1.82 Å data to $R$ and $R_{free}$ values of 0.1587 and 0.1817, respectively, with 96.86% of residues within the favoured regions of the Ramachandran plot (0 outliers), clashscore of 0.00 and overall MolProbity score of 0.69 (*Chen et al., 2010*). The final SUN1-KASH1 model was analysed using the *Online_DPI* webserver (http://cluster.physics.iisc.ernet.in/dpi) to determine a Cruikshank diffraction precision index (DPI) of 0.06 Å (*Kumar et al., 2015*).

## Size-exclusion chromatography multi-angle light scattering (SEC-MALS)

The absolute molar masses of protein samples and complexes were determined by size-exclusion chromatography multi-angle light scattering (SEC-MALS). Protein samples at >1 mg/ml (unless oth-erwise states) were loaded onto a Superdex 200 Increase 10/300 GL size exclusion chromatography column (GE Healthcare) in 20 mM Tris pH 8.0, 150 mM KCl, 2 mM DTT, at 0.5 ml/min using an ÄKTA Pure (GE Healthcare). The column outlet was fed into a DAWN HELEOS II MALS detector (Wyatt Technology), followed by an Optilab T-rEX differential refractometer (Wyatt Technology). Light scat-tering and differential refractive index data were collected and analysed using ASTRA six software (Wyatt Technology). Molecular weights and estimated errors were calculated across eluted peaks by extrapolation from Zimm plots using a dn/dc value of 0.1850 ml/g. SEC-MALS data are presented as differential refractive index (dRI) profiles with fitted molecular weights ($M_W$) plotted across elution peaks.

## Spectrophotometric determination of zinc content

The presence of zinc in protein samples was determined through a spectrophotometric method using the metallochromic indicator 4-(2-pyridylazo) resorcinol (PAR) (*Säbel et al., 2009*). Protein samples at 90–200 μM, corresponding to SUN1-KASH4 wild-type and CC381/382SS, and a wild-type sample that had been treated with EDTA (at a 10-fold molar excess relative to protein concentration) prior to gel-filtration, were digested with 0.6 μg/μl proteinase K (NEB) at 60°C for 1 hr. Of the super-natant, 10 μl of each protein digestion was added to 80 μl of 50 μM 4-(2-pyridylazo)-resorcinol (PAR) in 20 mM Tris, pH 8.0, 150 mM KCl, incubated for 5 min at room temperature, and UV absorbance spectra were recorded between 600 and 300 nm (Varian Cary 60 spectrophotometer). Zinc concen-trations were estimated from the ratio between absorbance at 492 and 414 nm, plotted on a line of best fit obtained from analysis of 0–100 μM zinc acetate standards.

## KASH-binding by SUN1 point mutants

The wild-type and individual point mutations I673E, F671E, and W676E of SUN1 and I579E of SUN2 (as His$_6$-GCN4 fusions) were co-expressed with KASH (as His$_6$-MBP fusion) as described above. Initial purification was performed by amylose affinity chromatography (NEB), relying on the residual affinity of SUN1/2 in cases when point mutations were disruptive. Resultant protein mixtures were analysed by ion exchange chromatography using HiTrap Q HP (GE Healthcare) and comparable samples from full elution profiles of wild-type and mutant proteins for each KASH binding-partner were analysed by SDS-PAGE. The entire elutions were then pooled, concentrated and analysed by size-exclusion chromatography on a Superdex 200 Increase 10/300 GL size exclusion chromatography column (GE Healthcare) in 20 mM Tris pH 8.0, 150 mM KCl, 2 mM DTT, at 0.5 ml/min using an ÄKTA Pure (GE Healthcare). Elution fractions of wild-type and mutant proteins for each KASH binding-partner were analysed by SDS-PAGE.

## Size-exclusion chromatography small-angle X-ray scattering (SEC-SAXS)

SEC-SAXS experiments were performed at beamline B21 of the Diamond Light Source synchrotron facility (Oxfordshire, UK). Protein samples at concentrations > 10 mg/ml were loaded onto a Super-dex 200 Increase 10/300 GL size exclusion chromatography column (GE Healthcare) in 20 mM Tris

pH 8.0, 150 mM KCl at 0.5 ml/min using an Agilent 1200 HPLC system. The column outlet was fed into the experimental cell, and SAXS data were recorded at 12.4 keV, detector distance 4.014 m, in 3.0 s frames. ScÅtter 3.0 (http://www.bioisis.net) was used to subtract, average the frames and carry out the Guinier analysis for the radius of gyration ($Rg$), and $P(r)$ distributions were fitted using *PRIMUS* (*Konarev et al., 2003*). Ab initio modelling was performed using *DAMMIF* (*Franke and Svergun, 2009*); 30 independent runs were performed in P1 and averaged. Crystal structures and models were fitted to experimental data using *CRYSOL* (*Svergun et al., 1995*). Normal mode analysis was used to model conformational flexibility for fitting to SAXS data using *SREFLEX* (*Panjkovich and Svergun, 2016*), and rigid body and flexible termini modelling was performed using *CORAL* (*Petoukhov et al., 2012*).

### Normal mode analysis of SUN1-KASH structures

Non-linear normal modes were calculated and visualised for SUN1-KASH 6:6 structures using the NOLB algorithm (*Hoffmann and Grudinin, 2017*) within the normal mode analysis SAMSON element (https://www.samson-connect.net).

### Protein sequence and structure analysis

Nesprin sequences were aligned and visualised using MUSCLE (*Madeira et al., 2019*) and Jalview (*Waterhouse et al., 2009*). Molecular structure images were generated using the PyMOL Molecular Graphics System, Version 2.3 Schrödinger, LLC.

## Acknowledgements

We thank Diamond Light Source and the staff of beamlines I04, I04-1 and B21 (proposals mx13587, mx18598 and sm15836), A Basle for assistance with X-ray crystallographic data collection, and J Dunce for critically reviewing the manuscript. ORD is a Sir Henry Dale Fellow jointly funded by the Wellcome Trust and Royal Society (Grant Number 104158/Z/14/Z).

## Additional information

### Funding

| Funder | Grant reference number | Author |
| --- | --- | --- |
| Wellcome Trust | 104158/Z/14/Z | Owen Richard Davies |
| Royal Society | 104158/Z/14/Z | Owen Richard Davies |

The funders had no role in study design, data collection and interpretation, or the decision to submit the work for publication.

### Author contributions

Manickam Gurusaran, Data curation, Validation, Investigation, Methodology, Project administration; Owen Richard Davies, Conceptualization, Formal analysis, Supervision, Funding acquisition, Visualization, Writing - original draft, Project administration, Writing - review and editing

### Author ORCIDs

Manickam Gurusaran https://orcid.org/0000-0002-6603-3118
Owen Richard Davies https://orcid.org/0000-0002-3806-5403

### Decision letter and Author response

Decision letter https://doi.org/10.7554/eLife.60175.sa1
Author response https://doi.org/10.7554/eLife.60175.sa2

## Additional files

### Supplementary files
• Transparent reporting form

### Data availability

Crystallographic structure factors and atomic coordinates have been deposited in the Protein Data Bank (PDB) under accession numbers 6R15, 6R16 and 6R2I, and raw diffraction data have been uploaded to https://proteindiffraction.org/. SEC-SAXS data have been deposited in the Small Angle Scattering Biological Data Bank (https://www.sasbdb.org/) under accession numbers SASDJC5, SASDJD5, SASDJE5 and SASDJF5. Uncropped gel images relating to Figures 4b and 5a are available in source data files.

The following datasets were generated:

| Author(s) | Year | Dataset title | Dataset URL | Database and Identifier |
|---|---|---|---|---|
| Gurusaran M, Davies OR | 2020 | Crystal structure of the SUN1-KASH1 6:6 complex | https://www.rcsb.org/structure/6R15 | RCSB Protein Data Bank, 6R15 |
| Gurusaran M, Davies OR | 2020 | Crystal structure of the SUN1-KASH4 6:6 complex | https://www.rcsb.org/structure/6R16 | RCSB Protein Data Bank, 6R16 |
| Gurusaran M, Davies OR | 2020 | Crystal structure of the SUN1-KASH5 6:6 complex | https://www.rcsb.org/structure/6R2I | RCSB Protein Data Bank, 6R2I |
| Gurusaran M, Davies OR | 2020 | LINC complex between the SUN domain of SUN1 and KASH domain of Nesprin-4 - SUN1-KASH4 6:6 complex | https://www.sasbdb.org/data/SASDJC5/ | Small Angle Scattering Biological Data Bank, SASDJC5 |
| Gurusaran M, Davies OR | 2020 | LINC complex between the SUN domain of SUN1 and KASH domain of KASH5 - SUN1-KASH5 6:6 complex | https://www.sasbdb.org/data/SASDJD5/ | Small Angle Scattering Biological Data Bank, SASDJD5 |
| Gurusaran M, Davies OR | 2020 | LINC complex between the SUN domain of SUN1 and KASH domain of Nesprin-1 - SUN1-KASH1 6:6 complex | https://www.sasbdb.org/data/SASDJE5/ | Small Angle Scattering Biological Data Bank, SASDJE5 |
| Gurusaran M, Davies OR | 2020 | SUN domain of SUN1 harbouring mutation I673E - monomer | https://www.sasbdb.org/data/SASDJF5/ | Small Angle Scattering Biological Data Bank, SASDJF5 |

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
