## [Decision Letter]

**Acceptance summary:**

This robust structural work provides new insights into the oligomeric state of the biochemically challenging SUN proteins, and results in the description of a new model for the physical link between the nucleoskeleton and cytoskeleton.

**Decision letter after peer review:**

Thank you for submitting your article "A molecular mechanism for LINC complex branching by structurally diverse SUN-KASH 6:6 assemblies" for consideration by *eLife*. Your article has been reviewed by three peer reviewers, one of whom is a member of our Board of Reviewing Editors, and the evaluation has been overseen by Cynthia Wolberger as the Senior Editor. The following individual involved in review of your submission has agreed to reveal their identity: Sophie Zinn-Justin (Reviewer #2).

The reviewers have discussed the reviews with one another and the Reviewing Editor has drafted this decision to help you prepare a revised submission.

Summary:

The current model of LINC complex organization features a SUN protein trimer that engages with up to three KASH peptides. Inner nuclear membrane SUN proteins, SUN1 and SUN2 in mammalian somatic cells, are anchored to nuclear components including the nuclear lamina, while outer nuclear membrane KASH proteins, which include Nesprin family members, bind a variety of cytoskeletal proteins. In this way, LINC complexes span the two nuclear membranes to mechanically couple nuclear and cytoplasmic structures. In this manuscript Gurusaran and Davies use biochemistry, SEC-MALS, X-ray crystallography, SAXS, and molecular modeling to probe the molecular underpinnings of several SUN-KASH interfaces, including SUN1 with KASH1, KASH4 and KASH5 taken from Nesprin-1, Nesprin-4 and KASH5, respectively. Taking on what has historically been a challenging biochemical system, they succeed in carrying out extensive biophysical analysis on the oligomeric states of these SUN-KASH complexes in solution. Applying these tools, the authors extend previous structural studies on the assembly of LINC complexes through the interactions of SUN and KASH domain proteins to reveal new insights. Specifically, these new biophysical studies support a model in which many LINC complex structures are likely exist in a 6:6 assembly mediated by two SUN trimers in a fashion that is influenced by the specific KASH peptide that is engaged. In this way, the somewhat atypical KASH proteins, KASH4 and KASH5 have a more significant role in stabilizing the head to head associations of SUN1 trimers than do the KASH domains from other Nesprins. Overall the reviewers found the experimental design and quality to be outstanding and robust while the new model for a force distribution network at the nuclear periphery facilitated by a branching network of LINC complexes is provocative. While all reviewers appreciated the structural biology and biophysics and were supportive of publication, there was some concern that it may fall short of providing a realistic view of LINC complex organization in vivo given the lack of supportive data.

1) Two of the three reviewers found the manuscript overly dismissive of the prior trimer model rather than seeing the work as an extension of previous studies (and found the discussion of the work in this context lacking). Indeed, prior work hypothesized that the trimer model could support a branching network depending on the configuration of SUN and KASH protomers and the cytoskeleton. For example, please see the work from Karakesisoglou and colleagues (Lu et al., 2012). A cautious interpretation by the prior structural biologists (Sosa et al.) given the relatively small buried surface area of the trimers (composed of SUN2) was not inappropriate (without data from solution studies) nor was it inconsistent with the observations here for the SUN1-KASH1 structure. Although the reviewers agree that the solution studies carried out by the authors are critical and provide new insight, they found it unnecessary to over-emphasize this as a major discrepancy, particularly as the authors do not analyze SUN2 in this study. Given all these points, such suggestions in the text should be toned down.

2) Aspects of the manuscript that are subject to alternative explanations that should be considered/discussed. Including:

a) The sentence "Thus, KASH-lids act as a hinge at the 6:6 interface, opening the linear crystal structure into an angled conformation that is seemingly more stable in solution" is misleading as it suggests that there is in solution only one structure in which the two trimers have a defined relative position different from that observed in the crystal. The reviewers felt that the manuscript lacks experimental evidence supporting the existence of a more stable structure in solution. Given this, the text should be edited or additional supportive data should be included.

b) The sentence "the asymmetrical hinged structure places all six KASH1 N-termini in favourable positions and orientations for their upstream transmembrane sequences to cross the outer nuclear membrane" was found to not be convincing. We encourage the authors to consider that some flexibility in the positioning of one trimer relatively to the other enables/facilitates the interaction between two trimeric complexes anchored in the same membrane.

3) Further clarification in the context of prior biological studies. Please clarify: in the arrangement of the SUN hexamer parallel to the ONM, individual KASH proteins would appear to be non-equivalent. This is compounded by the fact that it is not clear how an individual KASH dimer engages with the two SUN trimers. There are basically two possibilities. In the first, a KASH dimer would engage with a single SUN trimer, with the two KASH peptides occupying adjacent binding sites. The alternative would be that KASH dimers would engage binding sites within head to head paired SUN trimers, in this way tying the trimers together. In the first scheme, however, there would be a single unoccupied KASH binding site within each trimer. This would require a KASH dimer to either span both SUN trimers (as in the second scheme) or one KASH peptide to remain unbound. Does the conformation therefore depend to a large extent on the SUN-KASH stochiometry in vivo, not to mention the activities of chaperones such as Torsin A? This should be discussed in the context of the fluorescence fluctuation spectroscopy studies by the Luxton lab which directly addressed the issue of SUN protein oligomerization in vivo (Hennen et al., 2018).

4) Homogeneity versus heterogeneity in LINC complexes and SUN1 versus SUN2. Two of the reviewers felt that discussing the possibility for differences between SUN1 and SUN2 containing LINC complexes was important, particularly given that this is a major difference between this and prior structural studies. For example, it is likely that within their respective cell types (inner hair cells and spermatocytes), Nesprin 4 and KASH 5 are the predominant KASH proteins. Certainly in IHCs, SUN1 appears to be the only SUN protein. Thus LINC complexes in these cells would appear to be quite homogenous. This may not be the case for somatic cells. Do the authors envisage that different KASH proteins may pair up indiscriminately with available SUN proteins. This is not a trivial issue, since Gundersen and colleagues suggest that SUN1 may be bound preferentially by Nesprins associated with the microtubule system. SUN2 in contrast, appears to have a preference for actin-associated Nesprins (Zhu, Antoku and Gundersen, 2017). Moreover, there is additional evidence that the two SUN proteins are not redundant and may indeed be antagonistic (see Thakar et al., 2017, May and Carroll, 2018) – could this relate to the different types of assemblies? Is there any evidence, then, that SUN1 and SUN2 might have differing abilities to form 6:6 versus 3:3 LINC complexes and might this explain the differing specializations of these two proteins? Might SUN1 and SUN2 compete for the oligomeric state that LINC complexes are displaying? In other words, is it possible that SUN2 does not engage in these type of oligomers? While the authors do not assess SUN2 here, given the prior studies might it simply be a different story?

5) Discussion and interpretation of the SAXS data. the authors show that the SAXS data fits perfectly well with X-ray structure of SUN1-KASH4. In the case of SUN1-KASH5, modeling the N-termini greatly improved the fitting. Why does this modeling specifically impact the SAXS curve of SUN1-KASH5 and not SUN1-KASH1/5? If a rigid body modeling was performed instead for SUN1-KASH5, would it also be possible to fit the SAXS data? In the case of SUN1-KASH1, what is the impact of modeling the N-termini on the SAXS curve? Could the authors discuss about the fact that some mobility might exist on the relative positioning of SUN1 vs. KASH5 and SUN1 vs. KASH1, which could account for the difference between the SAXS and X-ray data?

6) The role of Zn. Is the presence of Zn atoms responsible for a more rigid interface in SUN1-KASH4? The zinc-mediated association of the Nesprin 4 6:6 complex is dependent upon the paired cysteines (C381 and C382) as well as H384. Has C381 also been suggested to form a disulphide with a cysteine in SUN1, comparable to that observed with KASH1 (from the Sosa paper)? By mutating the cysteines to serine, the Zn coordination is lost. Does this render the complex more similar to the "looser" structure observed with KASH1? Does treatment with EDTA/mutation of the relevant KASH4 cysteines disrupt the SUN1-KASH4 assembly?

Revisions expected in follow-up work:

1) Given that the most substantial criticism focused on the biological relevance of the 6:6 assembly in vivo, the reviewers wished to emphasize that the impact of the work would be greater if the authors provided some supportive data from cells focusing on how mutations that abrogate the identified interface necessary for 6:6 but not 3:3 assemblies affects LINC complex structures/function. Likewise, in vivo data to support the importance of Zn binding would strengthen the impact of this unexpected component. However, particularly given current constraints, the reviewers found that the compelling nature of the biochemical and biophysical studies was sufficient to warrant publication provided editing of the text according to the major points, above.

---

## [Author Response]

Revisions for this paper:1) Two of the three reviewers found the manuscript overly dismissive of the prior trimer model rather than seeing the work as an extension of previous studies (and found the discussion of the work in this context lacking). Indeed, prior work hypothesized that the trimer model could support a branching network depending on the configuration of SUN and KASH protomers and the cytoskeleton. For example, please see the work from Karakesisoglou and colleagues (Lu et al., 2012). A cautious interpretation by the prior structural biologists (Sosa et al.) given the relatively small buried surface area of the trimers (composed of SUN2) was not inappropriate (without data from solution studies) nor was it inconsistent with the observations here for the SUN1-KASH1 structure. Although the reviewers agree that the solution studies carried out by the authors are critical and provide new insight, they found it unnecessary to over-emphasize this as a major discrepancy, particularly as the authors do not analyze SUN2 in this study. Given all these points, such suggestions in the text should be toned down.

We agree with the reviewers that our work is an extension of previous studies, and similarly it was entirely appropriate for authors of the previous work to assume that the structures are 3:3 given the relatively small buried area of KASH1/2 structures and the lack of solution data. We have edited the manuscript throughout all sections to clarify these points (Abstract, Introduction, Results and Discussion).

Similarly, we agree that the idea of higher-order assembly had been raised numerous times in literature. We have expanded our discussion of this in the edited manuscript, including a description of how these ideas may integrate with our 6:6 assembly model to form a highly branched LINC complex network (Introduction and Discussion).

In response to the point that we had not analysed SUN2 in this study, we have now completed this analysis, confirming that it also forms 6:6 hetero-oligomers, with some notable differences (please see response number 4, below). Nevertheless, we are of the opinion that the above comments and responses are unaffected by these new findings.

2) Aspects of the manuscript that are subject to alternative explanations that should be considered/discussed. Including:a) The sentence "Thus, KASH-lids act as a hinge at the 6:6 interface, opening the linear crystal structure into an angled conformation that is seemingly more stable in solution" is misleading as it suggests that there is in solution only one structure in which the two trimers have a defined relative position different from that observed in the crystal. The reviewers felt that the manuscript lacks experimental evidence supporting the existence of a more stable structure in solution. Given this, the text should be edited or additional supportive data should be included.

Our intended meaning was entirely in agreement with the reviewer’s interpretation – that hinge motion would support the formation of a range of angled structures, and that angled structures (but not one definitive conformation) are seemingly more stable in solution than the linear conformation of the crystal structure.

b) The sentence "the asymmetrical hinged structure places all six KASH1 N-termini in favourable positions and orientations for their upstream transmembrane sequences to cross the outer nuclear membrane" was found to not be convincing. We encourage the authors to consider that some flexibility in the positioning of one trimer relatively to the other enables/facilitates the interaction between two trimeric complexes anchored in the same membrane.

Similarly, our intended meaning was that a range of angled conformations would allow the positioning of KASH molecules in appropriate orientations relative to the outer nuclear membrane. Further, we strongly agree that flexibility between opposing trimers would greatly facilitate assembly at the membrane.

To address the above points, we have extensively edited the relevant sections of the Results and Discussion to clarify our intended meaning. Specifically, we have explicitly stated that we envisage the solution state to be a continuous range of angled conformations that include but are not limited to the 60° structure determined by SAXS analysis. Further, we develop the idea of flexibility through hinge-like motion as a means of generating distinct LINC structures in response to the combined local environmental factors of tension force magnitude/direction, steric constraints and membrane structure.

3) Further clarification in the context of prior biological studies. Please clarify: in the arrangement of the SUN hexamer parallel to the ONM, individual KASH proteins would appear to be non-equivalent. This is compounded by the fact that it is not clear how an individual KASH dimer engages with the two SUN trimers. There are basically two possibilities. In the first, a KASH dimer would engage with a single SUN trimer, with the two KASH peptides occupying adjacent binding sites. The alternative would be that KASH dimers would engage binding sites within head to head paired SUN trimers, in this way tying the trimers together. In the first scheme, however, there would be a single unoccupied KASH binding site within each trimer. This would require a KASH dimer to either span both SUN trimers (as in the second scheme) or one KASH peptide to remain unbound. Does the conformation therefore depend to a large extent on the SUN-KASH stochiometry in vivo, not to mention the activities of chaperones such as Torsin A? This should be discussed in the context of the fluorescence fluctuation spectroscopy studies by the Luxton lab which directly addressed the issue of SUN protein oligomerization in vivo (Hennen et al., 2018).

The reviewer highlights an important point regarding the mode of interaction between SUN trimers and KASH dimers, and we agree with the alternative models proposed. We find that SUN1 complexes are extremely stable and we have never observed species with reduced occupancy. Thus, we think it is extremely unlikely that each KASH dimer would bind to a SUN trimer, leaving two binding sites unoccupied. Instead, we strongly favour the model in which each KASH dimer spans both SUN trimers, in which all KASH molecules interact symmetrically. It is notable that this conundrum is solved only by the 6:6 stoichiometry as it would not be possible to form a 3:3 complex with full occupancy in vivo (other than with asymmetric binding in which half of KASH molecules are split between different complexes). We have added a discussion of this point in the section in which we describe how variation in oligomer state along the SUN-KASH axis can provide branching events – in this case, the dimer-trimer interface provides a branching event across the outer nuclear membrane (Discussion).

Given the stability of SUN1 complexes, our observations of only full-occupancy complexes and the nature of the structure, we think it is unlikely that the in vivo stoichiometry would affect the nature of SUN-KASH binding. Instead, we suggest that the limiting component would determine the amount of 6:6 complex formed, with the component in excess remaining unbound. An excess of SUN proteins would likely result in unbound molecules adopting their “autoinhibited” conformation in which trimerization and KASH-binding of the SUN domain is blocked. This may be important in establishing discrete pools of assembled and unassembled SUN molecules in which autoinhibited SUN molecules are unable to compete for KASH-binding which would weaken established LINC assemblies. We have integrated this point into our wider discussion of autoinhibition and alternative conformations (Discussion).

We agree that proteins chaperones (alongside enzymatic modification, protein interactions and chemical conditions) may play important roles in guiding LINC complex assembly (including overcoming autoinhibition in a timely manner). We have included a description of this and highlight some potential mechanisms in the Discussion.

The fluorescence fluctuation spectroscopy studies are important in understanding the oligomeric assembly of the SUN’s luminal regions within the context of the nuclear envelope. It is important to point out that individual KASH domains and isolated SUN domains remained monomeric in these experiments, suggesting that the expressed proteins did not undergo SUN-KASH complex formation with endogenous proteins of the nuclear envelope. Thus, the oligomeric states observed do not determine the stoichiometry of SUN-KASH complexes but do describe the oligomeric assembly of the coiled-coils of SUN’s luminal region within the nuclear envelope. Indeed, the authors state that full-length SUN proteins were highly immobile, which is why they had to analyse luminal fragments that proved to be mobile and thereby suitable for FFS analysis. We would suggest that immobility may be a consequence of LINC network assembly upon KASH-binding. Nevertheless, the determined oligomeric states of the SUN luminal regions greatly add to our understanding – these were observed to be trimers, with additional higher order structures for SUN1. These findings are consistent with SUN-KASH complexes establishing head-to-head assembly of SUN trimers, with possible additional branching between SUN1 trimers. We have added a discussion of this to the manuscript (Introduction and Discussion).

4) Homogeneity versus heterogeneity in LINC complexes and SUN1 versus SUN2. Two of the reviewers felt that discussing the possibility for differences between SUN1 and SUN2 containing LINC complexes was important, particularly given that this is a major difference between this and prior structural studies. For example, it is likely that within their respective cell types (inner hair cells and spermatocytes), Nesprin 4 and KASH 5 are the predominant KASH proteins. Certainly in IHCs, SUN1 appears to be the only SUN protein. Thus LINC complexes in these cells would appear to be quite homogenous. This may not be the case for somatic cells. Do the authors envisage that different KASH proteins may pair up indiscriminately with available SUN proteins. This is not a trivial issue, since Gundersen and colleagues suggest that SUN1 may be bound preferentially by Nesprins associated with the microtubule system. SUN2 in contrast, appears to have a preference for actin-associated Nesprins (Zhu, Antoku and Gundersen, 2017). Moreover, there is additional evidence that the two SUN proteins are not redundant and may indeed be antagonistic (see Thakar et al., 2017, May and Carroll, 2018) – could this relate to the different types of assemblies? Is there any evidence, then, that SUN1 and SUN2 might have differing abilities to form 6:6 versus 3:3 LINC complexes and might this explain the differing specializations of these two proteins? Might SUN1 and SUN2 compete for the oligomeric state that LINC complexes are displaying? In other words, is it possible that SUN2 does not engage in these type of oligomers? While the authors do not assess SUN2 here, given the prior studies might it simply be a different story?

The reviewers raise an important question, which we addressed by purifying and performing gel filtration and SEC-MALS analysis of SUN-KASH complexes formed between SUN2 and KASH1/4/5. We find that SUN2-KASH4 complexes are 6:6 hetero-oligomers of similar stability to SUN1 complexes. Our analysis of SUN2-KASH1/5 complexes was complicated by their substantially reduced stability in comparison with the highly stable SUN1 complexes, in which they readily dissociated during purification and SEC-MALS analysis. Nevertheless, SEC-MALS analysis revealed that their dissociating complex species had molecular weight ranges that include and exceed (by approximately double) the theoretical molecular weights of 6:6 complexes, and are far in excess of the theoretical molecular weights of 3:3 complexes. These findings suggest that SUN2-KASH1/5 are dissociating structures of 6:6 and larger hetero-oligomers. We improved the clarity of SUN2-KASH1 data by using SUN2 mutation C705A, which targets a non-structural cysteine residue to prevent it from forming disulphide bonds. This mutation blocked the higher-order assembly of SUN2-KASH1, and demonstrated a clear (but still dissociating) 6:6 hetero-oligomer. We further tested SUN2 mutation I579E, a direct copy of SUN1 mutation I673E, which targets the isoleucine residue that solely mediates interactions across the 6:6 interface in the SUN-KASH1 structures, and would be entirely solvent-exposed in a 3:3 complex. The SUN2 I579E mutation completely disrupted SUN2-KASH1 complex formation, in precisely the same way as SUN1 mutation disrupted SUN1-KASH1 complex formation. Thus, we can conclude that the SUN2-KASH1 is entirely dependent on the 6:6 interface for stability of the complex. Together, our data demonstrate that SUN2-KASH complexes are 6:6 hetero-oligomers (and larger structures) and that the 6:6 interface is essential for SUN2-KASH1 complex formation. Hence, our findings and conclusions regarding 6:6 assembly apply to LINC complexes formed of SUN1 and SUN2 proteins. We have included these data as new Figure 5 and inserted a discussion of these findings in the text (Results and Discussion).

Our findings that SUN2-KASH1/5 form higher-order structures and are substantially less stable that SUN1-KASH1/5 may explain some of the observed functional differences between SUN1 and SUN2. Specifically, reduced stability could favour a higher rate of turnover of a LINC complex assembly, whilst higher-order structures could establish different branched LINC complex architectures. We have included a discussion of these ideas, how they may explain some observed differences between SUN proteins, and wider means by which LINC complexes may form distinct structures to achieve specialised functions (Discussion).

5) Discussion and interpretation of the SAXS data. the authors show that the SAXS data fits perfectly well with X-ray structure of SUN1-KASH4. In the case of SUN1-KASH5, modeling the N-termini greatly improved the fitting. Why does this modeling specifically impact the SAXS curve of SUN1-KASH5 and not SUN1-KASH1/5? If a rigid body modeling was performed instead for SUN1-KASH5, would it also be possible to fit the SAXS data? In the case of SUN1-KASH1, what is the impact of modeling the N-termini on the SAXS curve? Could the authors discuss about the fact that some mobility might exist on the relative positioning of SUN1 vs. KASH5 and SUN1 vs. KASH1, which could account for the difference between the SAXS and X-ray data?

As SAXS is a relatively low-resolution technique, it is important to prevent over-fitting of data. In essence, given a sufficiently large number of parameters, it would be possible to fit data with any model, but this would poorly reflect the true structure. A prudent approach is to start with simplest explanation of the data (minimum number of parameters) and only increase the complexity and number of parameters if necessary and in increments (utilising Occam’s razor). In this case, we started with the crystal structure, then added unstructured residues that are missing from the structure but are definitely present in the sample, and only if this failed did we explore alternative conformations. SUN1-KASH4/5 could have been fitted with rigid body models, but this would have added unnecessary potential for artefactual results given that the data could be adequately explained by a much simpler model. In contrast, SUN1-KASH1 could not be explained by the simple model, demonstrating that large-scale structural alterations must have occurred. The reason for this can be seen in the data as the real-space plot shows that the maximum dimension of SUN1-KASH1 is slightly shorter than SUN1-KASH4/5 (Figure 6B). This shortening of the molecule could never be modelled by adding termini and can only be explained by its predicted angulation. Importantly, we think that all three complexes undergo hinge-like motion through their 6:6 interface (as demonstrated by normal mode analysis – Figure 7) and thereby exist in a range of angled conformations in solution. For SUN1-KASH1, there must be a large proportion of highly angled conformations that dominant the ensemble SAXS data. In contrast, the proportion and angulation of SUN1-KASH4/5 structures is likely lower, meaning that their ensemble SAXS data more closer resemble the linear conformation. We have edited our discussions of the SAXS data, normal modes analysis and interpretation of hinge-like motion to clarify these points (Results and Discussion).

6) The role of Zn. Is the presence of Zn atoms responsible for a more rigid interface in SUN1-KASH4? The zinc-mediated association of the Nesprin 4 6:6 complex is dependent upon the paired cysteines (C381 and C382) as well as H384. Has C381 also been suggested to form a disulphide with a cysteine in SUN1, comparable to that observed with KASH1 (from the Sosa paper)? By mutating the cysteines to serine, the Zn coordination is lost. Does this render the complex more similar to the "looser" structure observed with KASH1? Does treatment with EDTA/mutation of the relevant KASH4 cysteines disrupt the SUN1-KASH4 assembly?

The zinc sites provide a distinct 6:6 interface for SUN1-KASH4, in which residues that are responsible for head-to-head assembly of SUN1-KASH1 are redundant, and the symmetrical nature of the sites likely accounts for a higher proportion of linear conformations in solution. It is notable that SUN2-KASH4 was the only SUN2 complex that retained the high binding affinity of the SUN1 complex, which is explained as the 6:6 interface is mediated solely by KASH4 zinc sites, whereas KASH1/5 complexes involve SUN residues at the interface, where sequences differences between SUN1 and SUN2 could explain the substantially reduced affinity of SUN2-KASH1/5. We have added this point to the Discussion.

The zinc ligand C381 was proposed by Sosa et al. to form a disulphide with SUN1, but this was based on a KASH domain alignment and the assumption that KASH4 would adopt the same structure as KASH1 (in which a cysteine at this position does form a disulphide with SUN1). We now know that this assumption was incorrect as KASH4 has a different conformation to KASH1 so their cysteine residues are non-equivalent.

We have analysed the oligomeric state of SUN1-KASH4 upon removal of bound zinc by EDTA-treatment, and find that it retains 6:6 structure and also forms a new 12:12 species, with no loss of complex stability. The retention of 6:6 assembly is simply explained by the structure reverting to the KASH1-like mode 6:6 interface, as correctly predicted by the reviewer. We suggest that the 12:12 structures likely consist of two 6:6 assemblies (with KASH1-like interfaces), interacting through KASH4 sequences that were liberated from zinc-sites. These interactions could include disulphide bond formation by newly exposed cysteine residues. We have added these data to the manuscript (Figure 3—figure supplement 2), and have included a description of how these conformational changes provide a potential regulatory mechanism upon change to zinc availability and sequestration within the nuclear envelope (Results and Discussion).

Revisions expected in follow-up work:1) Given that the most substantial criticism focused on the biological relevance of the 6:6 assembly in vivo, the reviewers wished to emphasize that the impact of the work would be greater if the authors provided some supportive data from cells focusing on how mutations that abrogate the identified interface necessary for 6:6 but not 3:3 assemblies affects LINC complex structures/function. Likewise, in vivo data to support the importance of Zn binding would strengthen the impact of this unexpected component. However, particularly given current constraints, the reviewers found that the compelling nature of the biochemical and biophysical studies was sufficient to warrant publication provided editing of the text according to the major points, above.

We agree with the reviewers that analysis of mutations that test the 6:6 assembly and zinc-binding in vivo would have enhanced the impact of the manuscript, and we had initiated these studies in 2019. However, as recognised by the reviewers, we have not been able to continue these studies owing to the impact of the pandemic. Nevertheless, we will reinitiate this work as soon as restrictions are lifted and will make our findings publicly available. We have added an acknowledgement of this limitation in the Discussion section.